# 3DID: Direct 3D Inverse Design for Aerodynamics with Physics-Aware Optimization

**Yuze Hao[1], Linchao Zhu[1,2], Yi Yang[1,2]***

[1] College of Computer Science and Technology, Zhejiang University
[2] The State Key Lab of Brain-Machine Intelligence, Zhejiang University

## Abstract

Inverse design aims to design the input variables of a physical system to optimize a specified objective function, typically formulated as a search or optimization problem. However, in 3D domains, the design space grows exponentially, rendering exhaustive grid-based searches infeasible. Recent advances in deep learning have accelerated inverse design by providing powerful generative priors and differentiable surrogate models. Nevertheless, current methods tend to approximate the 3D design space using 2D projections or fine-tune existing 3D shapes. These approaches sacrifice volumetric detail and constrain design exploration, preventing true 3D design from scratch. In this paper, we propose a **3D Inverse Design** (**3DID**) framework that directly navigates the 3D design space by coupling a continuous latent representation with a physics-aware optimization strategy. We first learn a unified physics–geometry embedding that compactly captures shape and physical field data in a continuous latent space. Then, we introduce a two-stage strategy to perform physics-aware optimization. In the first stage, a gradient-guided diffusion sampler explores the global latent manifold. In the second stage, an objective-driven, topology-preserving refinement further sculpts each candidate toward the target objective. This enables **3DID** to generate high-fidelity 3D geometries, outperforming existing methods in both solution quality and design versatility.

## 1 Introduction

Inverse design seeks to identify the initial variables of a physical system that, under given constraints, optimizes a specified objective function. This fundamental challenge occurs across many scientific and engineering disciplines, such as materials science, mechanical engineering, aerospace design, and supports applications ranging from automotive body shaping [1] and nano-photonic device engineering [2] to mechanical materials design [3, 4] and physics detector development [5].

Despite its broad impact, efficiently exploring the design space toward a target objective presents significant challenges. First, inverse design must contend with the inherent complexity of simulating physical systems for evaluation. These simulations are often nonlinear and high-dimensional, requiring fine discretizations that dominate computational resources [6, 7]. Second, the design landscape is extremely large, inherently nonconvex, and riddled with local minima, making exhaustive global search infeasible [8, 9]. In 3D domains, where inverse design usually involves direct geometry optimization, the number of degrees of freedom grows exponentially [10]. This rapid growth in geometric complexity drives up simulation expense and intensifies the search challenge.

To tackle inverse design in the 3D domain, various techniques have been proposed [1, 11, 12], yet they fall short of addressing the above challenges. Traditional approaches such as adjoint-based gradient methods [13, 14, 15] and Bayesian optimization [16, 17] provide broad applicability but depend on

---

*Corresponding author.

39th Conference on Neural Information Processing Systems (NeurIPS 2025).

repeated high-fidelity simulations that incur prohibitive computational cost. With recent advances in deep learning, pretrained surrogate models [18, 19, 20, 21] can efficiently approximate the forward physical process and support end-to-end backpropagation to update design variables, speeding up convergence by orders of magnitude. However, many prior methods adopt two simplifications. One replaces the 3D design with 2D proxies [1, 11] (multi-view renderings or silhouettes), which removes geometry information. The other requires an initial geometry as the starting point for subsequent refinement [22, 23, 24]. In practice, both assumptions restrict design exploration and hinder a thorough search of complex 3D design spaces (see Fig. 1). As a result, they cannot support the full exploration of complex three-dimensional design spaces, limiting coverage to a narrow subset of feasible geometries.

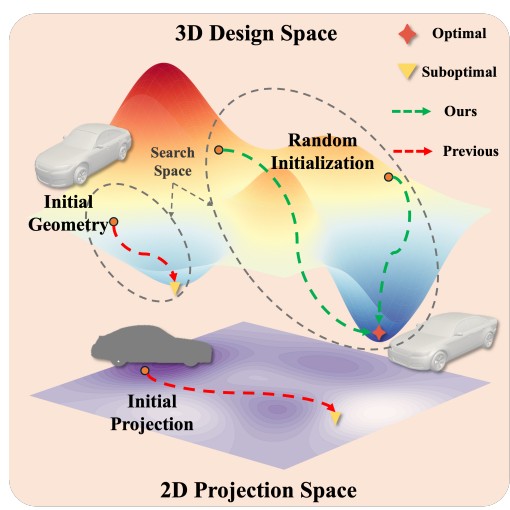

Figure 1: **Motivation of 3DID.** Existing 3D inverse-design methods either rely on reduced-dimensional representations (2D projections or fixed parameterizations) that constrain design freedom, or require an initial geometry as a starting point for local refinement, which highly constrains the search space. In contrast, 3DID overcomes these limitations by directly exploring the full 3D design space from random initialization.

We identify two primary challenges in 3D inverse design. 1) The high dimensionality of 3D physics-geometry-coupled spaces impedes exploration. Inverse design must simultaneously optimize geometric structures while accurately evaluating their resulting physical properties. This coupling, combined with the continuous high-resolution nature of both shape and physical fields, makes the direct 3D exploration extraordinarily difficult. 2) The lack of optimization strategies that balance the exploration–validity trade-off. Refining a baseline geometry with a surrogate model ensures constraint compliance and design validity, but it confines the search to a local neighborhood and can introduce adversarial artifacts when driven too far [25, 26]. On the other hand, sampling candidates with a generative model offers broader exploration yet leaves results vulnerable to biases in the training data [27]. Consequently, samples stay tethered to the prior and tend to imitate prevalent patterns rather than pushing toward novel optima.

To address these challenges, we introduce **3DID**, a 3D inverse-design framework that explores the design space without relying on simplified parameterizations or predefined shapes. Rather than directly searching the prohibitively large, continuous physics-geometry-coupled space, we first learn a continuous physics–geometry unified latent representation. This compact embedding preserves fine-grained shape and physical field variations while dramatically reducing both dimensionality and computational cost, thereby overcoming the dual obstacles of large-scale shape optimization and physics-aware simulation. Building on this latent space, we then deploy a two-stage optimization pipeline to tackle the exploration–validity trade-off. It begins with a gradient-guided diffusion sampler that traverses the manifold from pure noise to generate diverse, physics-informed candidates by steering sampling toward high-performance regions using objective gradients. Each candidate then undergoes topology-preserving optimization, which further improves objective performance under strict mesh-quality and connectivity constraints, ensuring geometric integrity and preventing adversarial artifacts. Together, these components enable 3DID to discover novel, high-fidelity 3D designs that reliably meet target objectives. In summary, our contributions are threefold:

(1) We propose a continuous latent embedding that jointly encodes detailed 3D geometry and high-fidelity physical fields, enabling an efficient, unified search within a compact latent manifold.

(2) We develop a two-stage optimization pipeline that begins with gradient-guided diffusion sampling for global exploration and followed by topology-preserving refinement, optimizing each candidate toward the desired objective while strictly maintaining structural integrity.

(3) We validate our 3DID framework on aerodynamic shape optimization, demonstrating that it consistently generates novel geometries whose superior performance is confirmed through surrogate evaluations and high-fidelity CFD simulations, significantly surpassing baseline methods.

## 2 Related work

### 2.1 Inverse Design

Compared with the forward PDE problem, which predicts the physical response of a given design using numerical solvers or learned surrogates [28, 29, 30, 31, 32, 33], inverse design seeks the design variables that achieve a target objective under engineering constraints [26, 18, 34]. Inverse design is a fundamental problem in many domains of science and engineering disciplines, including mechanical engineering [35, 36], material science [37, 38, 39], chemical engineering [40], medical engineering [41], and aerospace engineering [22, 6, 42]. Classical approaches typically combine high fidelity physics solvers with sampling-based optimization methods such as the Cross Entropy Method [43] or Gaussian-process model with Bayesian optimization [44] to explore the design space. With the advent of differentiable simulators, inverse design can be posed directly as a gradient-based optimization problem [41, 45]. More recently, deep learning driven methods have shown great promise by learning surrogate models that approximate forward physics and allow end-to-end backpropagation [18, 19]. Furthermore, generative models, including variational autoencoders [46], GAN [47, 48] and diffusion models [26, 49] have been applied to inverse design. While prior methods mainly excel in 2D or low-dimensional settings, we propose a framework that directly navigates the full 3D design space via physics-aware optimization.

### 2.2 Aerodynamic Shape Optimization

Aerodynamic shape optimization is a classical inverse design task that seeks geometries minimizing drag while satisfying constraints on lift, stability, and other performance criteria [50, 51, 52, 53]. Generally, effective optimization critically depends on two key components: shape representation and optimization strategy. Traditional approaches typically employ simplified, low-dimensional representations such as 2D projections [11, 54, 1] or spline-based parameterizations [55, 56, 57] to reduce dimensionality and computational costs. Optimization is then performed using gradient-based adjoint solvers for efficient local refinement [24, 58, 59]. Additionally, to accelerate convergence, many methods optimize from a pre-selected baseline geometry [22, 23, 24]. In contrast, we propose a guided diffusion model over a latent shape representation, enabling the design of unconstrained 3D geometries directly from noise, without relying on initial shapes or 2D profiles.

## 3 Methods

In this section, we first formalize the 3D inverse design problem (Section 3.1). We then introduce our physics–geometry unified representation (Section 3.2), describe the gradient-guided diffusion sampling process (Section 3.3), and detail the topology-preserving refinement stage (Section 3.4). Finally, we provide our implementation details (Section 3.5).

### 3.1 Problem Formulation

We consider the problem of 3D inverse design, where the goal is to identify a solid input geometry $M$ for a physical system that optimizes specified performance objectives while satisfying geometric constraints. Formally, let $M \subset \mathbb{R}^3$ denote a solid geometry, and let $\mathcal{F}(M)$ be the corresponding steady-state physical field (e.g., pressure or temperature distribution) governed by a partial differential equation (PDE) or an ordinary differential equation (ODE). We define the design objective as:

$$\mathcal{J}(M) := \mathcal{J}\big(M, \mathcal{F}(M)\big), \tag{1}$$

which may measure quantities such as drag, lift, and structural compliance. Specifically, $\mathcal{J}$ depends on $M$ in two ways: implicitly, via the resulting physical field $\mathcal{F}(M)$ on which $\mathcal{J}$ is evaluated, and explicitly, via direct geometric properties defined on $M$. The classical inverse design aims to solve:

$$M^* = \arg\min_{M} \ \mathcal{J}\big(M, \mathcal{F}(M)\big)$$
$$\text{s.t.} \ C\big(M, \mathcal{F}(M)\big) \leq 0, \tag{2}$$

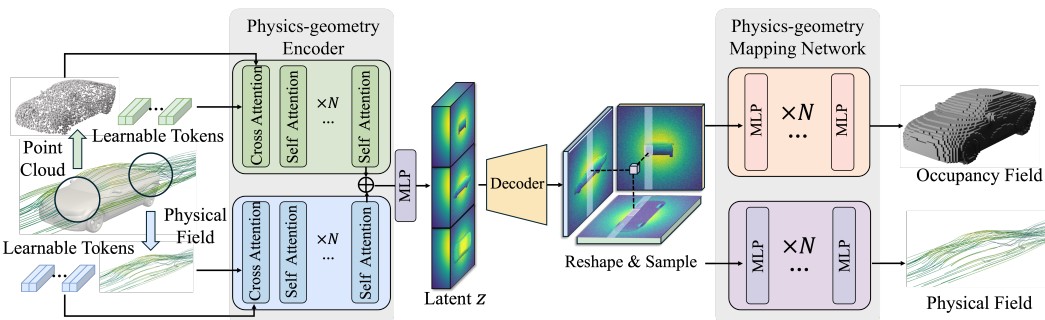

Figure 2: **The overview of PG-VAE.** We use transformers to encode the design geometry and its associated physical field, along with learnable tokens, into a compact triplane latent representation $z$. A decoder then upsamples the latent $z$ into high-resolution triplane feature maps, which can be reshaped into three orthogonal planes. Finally, a physics–geometry mapping network is applied to reconstruct both the occupancy field and the corresponding physical field from these feature maps.

where the solution $M^*$ is the geometry that minimizes the performance objective, $C$ aggregates design constraints, such as volume, manufacturability, and boundary conditions.

## 3.2 Physics-Geometry Unified Representation

To compactly encode 3D geometry and its physical field, we adopt a triplane representation learned via our Physics-Geometry VAE (PG-VAE). As shown in Figure 2, it includes: (1) A *physics–geometry encoder* that maps input geometry and physical field into a latent code. (2) A *latent-to-triplane decoder* that reconstructs triplane feature maps from the latent code. (3) A *physics–geometry mapping network* that recovers the 3D geometry and physical field from the triplane.

**Physics-Geometry Encoder.** The physics–geometry encoder comprises two parallel branches: one for processing raw 3D geometry, and the other for encoding the associated physical field. Inspired by [60], each branch uses learnable tokens to capture both local and global context. For the geometry branch, given uniformly sampled point clouds $P_{geo} \in \mathbb{R}^{N_g \times C_g}$ from the geometry, where $N_g$ is the number of points and $C_g$ represents features including normalized positions and surface normals, we utilize Fourier features [61] to encode positional structure. Then, a set of learnable tokens $e_g \in \mathbb{R}^{(3 \times r \times r) \times d_e}$ queries information from these points via cross-attention and self-attention layers, resulting in geometry latent tokens $z_g \in \mathbb{R}^{(3 \times r \times r) \times d_z}$. The physical-field branch follows the same structure, processing uniformly sampled physical-field points $P_{phy} \in \mathbb{R}^{N_p \times C_p}$, where $N_p$ is the number of points and $C_p$ is the dimension of physical-field features. Learnable tokens $e_p \in \mathbb{R}^{(3 \times r \times r) \times d_e}$ undergo similar attention layers to produce physical-field latents $z_p \in \mathbb{R}^{(3 \times r \times r) \times d_z}$. Finally, geometry and physical tokens are concatenated and passed through MLP layers to form the unified latent representation $z = \text{MLP}(\text{Concat}(z_g, z_p))$, where $z \in \mathbb{R}^{(3 \times r \times r) \times d_z}$.

**Latent-to-Triplane Decoder.** After obtaining the unified physics–geometry latent representation, we apply a decoder to formulate the triplane representation. Prior to decoding, we reshape the latent tokens by vertically concatenating three planes, yielding the reshaped latent tensor $z \in \mathbb{R}^{r \times (3 \times r) \times d_z}$, following [62]. Subsequently, the latent tensor is passed through a series of convolutional layers for upsampling. The output is then reshaped into the final triplane features $T_{xy}, T_{xz}, T_{yz} \in \mathbb{R}^{R \times R \times d_t}$, where $R$ denotes the resolution of each plane and $d_t$ is the feature dimension per pixel.

**Physics-Geometry Mapping Network.** The mapping network serves to reconstruct 3D geometry and the associated physical field from the learned triplane representation. We utilize two parallel MLP branches: one for predicting geometric occupancy and one for estimating the physical field. Given a query point $q \in \mathbb{R}^3$, we project it onto the three orthogonal planes and extract features. The aggregated feature is computed as $t_q = T_{xy}(q_{xy}) + T_{xz}(q_{xz}) + T_{yz}(q_{yz})$. where $q_{xy}, q_{xz}, q_{yz}$ denote the 2D projections of $q$ onto the respective planes. The aggregated feature $t_q$ is then fed into the MLP branches to predict the occupancy field and physical field values at the corresponding point $q$.

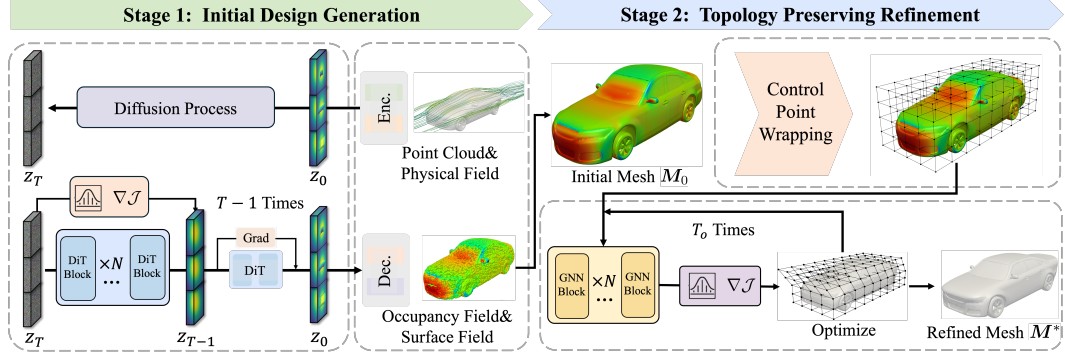

Figure 3: **The optimization framework of 3DID.** Starting from noise, we guide diffusion using objective gradients to steer the latent toward high-performance regions. The decoded triplane then yields an initial mesh $M_0$ and its surface physical field $\varphi$, which is then refined via backpropagation over a free-form deformation lattice to improve performance while preserving topology.

**End-to-End Training.** Our VAE model is trained end-to-end to jointly reconstruct 3D geometry and the associated physical field. For occupancy field reconstruction, we employ the Binary Cross-Entropy (BCE) loss $\mathcal{L}_{\text{BCE}}$ to supervise the predicted occupancy. To reconstruct the physical field, we utilize the Mean Squared Error (MSE) loss $\mathcal{L}_{\text{MSE}}$. Additionally, we incorporate a KL divergence loss $\mathcal{L}_{\text{KL}}$ to regularize the latent space. Overall, our training loss can be formulated as:

$$\mathcal{L}_{\text{PG-VAE}} = \lambda_{\text{BCE}}\mathcal{L}_{\text{BCE}} + \lambda_{\text{MSE}}\mathcal{L}_{\text{MSE}} + \lambda_{\text{KL}}\mathcal{L}_{\text{KL}}, \tag{3}$$

where $\lambda_{\text{BCE}}, \lambda_{\text{MSE}}, \lambda_{\text{KL}}$ are weighting coefficients.

## 3.3 Objective Guided Diffusion

Once the PG-VAE is trained, it provides a compact, expressive latent code $z$ that jointly captures 3D geometry and its physical field. We then train a diffusion model [63, 64] on these latents, enabling direct generation of samples on the learned manifold from pure noise. To drive inverse design, we inject gradients of the task objective $\mathcal{J}$ into the diffusion sampling, as shown in Figure 3.

In standard diffusion sampling, each denoising step predicts noise via the learned score function:

$$\epsilon_\phi(z_t, t) = -\sqrt{1 - \alpha_t}\, \nabla_{z_t} \log p(z_t), \tag{4}$$

where $\nabla_{z_t} \log p(z_t)$ denotes the score function, *i.e.*, the gradient of the log-probability density of the latent variable $z_t$. By iteratively denoising, the model guides samples toward high-probability regions of the data manifold. In our case, we need to consider not only guiding the noise towards the feasible data manifold, but we also need to incorporate optimization of $\mathcal{J}$ during the sampling. Therefore, inspired by [65, 66], we replace the unconditional score with the conditional score $\nabla_{z_t} \log p(z_t \mid \mathcal{J})$. By Bayes' rule, we can derive:

$$\nabla_{z_t} \log p(z_t \mid \mathcal{J}) \propto \nabla_{z_t} \log p(z_t) + \nabla_{z_t} \log p(\mathcal{J} \mid z_t). \tag{5}$$

Here, $\nabla_{z_t} \log p(z_t)$ corresponds to the standard score function learned by the diffusion model, while $\nabla_{z_t} \log p(\mathcal{J} \mid z_t)$ acts as an additional guidance term that incorporates the influence of the design objective. Since $\nabla_{z_t} \log p(\mathcal{J} \mid z_t)$ is unknown, we approximate it by:

$$\nabla_{z_t} \log p(\mathcal{J} \mid z_t) \simeq \nabla_{z_t} \log p\big(\mathcal{J} \mid \hat{z}_0(z_t)\big) \propto -\nabla_{z_t} \mathcal{J}(\hat{z}_0(z_t)), \tag{6}$$

where $\hat{z}_0(z_t)$ denotes the estimate of the clean latent code given the noisy latent $z_t$, following [66]. Accordingly, we adjust the predicted noise to incorporate the influence of the design objective $\mathcal{J}$, resulting in the guided noise prediction:

$$\epsilon'_\phi(z_t, t) = \epsilon_\phi(z_t, t) + \gamma \nabla_{z_t} \mathcal{J}, \tag{7}$$

where $\epsilon'_\phi$ is the modified noise prediction, and $\gamma$ is a scaling coefficient that controls the strength of the guidance. This objective-aware adjustment steers the sampling trajectory toward latent regions that both conform to the learned data distribution and advance the target design objective $\mathcal{J}$.

## 3.4 Topology-Preserving Refinement

After objective-guided diffusion, we obtain an optimized latent code $z^*$, which is reshaped into triplane feature maps and decoded by the physics–geometry mapping network to generate an initial 3D mesh $\boldsymbol{M}_0$ with vertex set $\boldsymbol{V}_0 = \{v_j\}_{j=1}^N$ and its associated physical field $\boldsymbol{\varphi} = \{\varphi_j\}_{j=1}^N$, as shown in Figure 3. Although guided by the design objective, the generated designs remain highly biased by the prior distribution of designs from the training data [27]. We introduce a topology-preserving refinement stage based on free-form deformation (FFD) [67, 68], controlled by gradient descent.

Specifically, we first wrap $\boldsymbol{M}_0$ in a 3D lattice of control points $\boldsymbol{C} = \{c_i\}_{i=1}^K$. These control points form a flexible control grid that allows smooth and structured adjustment of the mesh shape while preserving its topology. The deformation of each vertex $v_j$ is computed as:

$$v'_j(\boldsymbol{C}) = \sum_{i=1}^K \mathcal{B}_i(v_j)\, c_i, \tag{8}$$

where $\mathcal{B}_i(v_j)$ denotes the $i$-th trivariate Bernstein basis function [68] evaluated at the normalized parametric coordinate of vertex $v_j$. These basis functions provide smooth, localized influence from the control points, enabling flexible yet coherent deformation across the mesh.

At the beginning of the refinement process, the control points are unmodified, so $v'_j(\boldsymbol{C}) = v_j$. The initial vertex–field pairs $\{(v'_j(\boldsymbol{C}), \varphi_j)\}$ are then fed into a pretrained GNN surrogate $f_{\text{GNN}}$ which estimates the current design objective based on the mesh geometry and physical attributes:

$$\hat{\mathcal{J}}(\boldsymbol{C}) = f_{\text{GNN}}\left((v'_j(\boldsymbol{C}), \varphi_j)_{j=1}^N\right). \tag{9}$$

With this differentiable surrogate model, we optimize the control points to improve the design objective. The overall refinement loss is defined as:

$$\mathcal{L}(\boldsymbol{C}) = \hat{\mathcal{J}}(\boldsymbol{C}) + \lambda_{\text{smooth}} \sum_{i=1}^K \|\Delta c_i\|^2 + \lambda_{\text{vol}} \sum_{\text{cells}} \left(\frac{V_{\text{def}}}{V_{\text{orig}}} - 1\right)^2, \tag{10}$$

where $\Delta c_i$ are control-point displacements, and the term weighted by $\lambda_{\text{smooth}}$ penalizes large displacements for smooth deformations, while the term weighted by $\lambda_{\text{vol}}$ penalizes cell-wise volume changes to ensure geometric consistency. Control points are updated via:

$$\boldsymbol{C}^{(t+1)} = \boldsymbol{C}^{(t)} - \eta \nabla_{\boldsymbol{C}} \mathcal{L}(\boldsymbol{C}^{(t)}), \tag{11}$$

We optimize using AdamW with a cosine-annealed learning rate $\eta$. Iteration continues until an iteration count $T_o$ is reached. The resulting vertices $\boldsymbol{V}^* = \{v_j^*\}$, obtained via the FFD mapping (Eq. 8), define the refined mesh $\boldsymbol{M}^*$, which preserves topology and improves target performance.

## 3.5 Implementation Details

To train our PG-VAE, we sample $N_g = N_p = 50{,}000$ points as input and use the physics–geometry encoder with one cross-attention layer and 8 self-attention layers with 12 heads and $d_z = 64$, plus $r = 64$ learnable tokens of dimension $d_e = 768$, yielding a latent code of $d_z = 32$. The decoder upsamples via one self-attention layer and five ResNet blocks [69] to a triplane with $R = 256$ and channel $d_t = 64$. Each branch's mapping network has five linear layers with a hidden dimension of 32. We train the VAE model with loss weights $\lambda_{\text{BCE}} = 10^{-3}$, $\lambda_{\text{MSE}} = 10^{-5}$, $\lambda_{\text{KL}} = 10^{-6}$. During training, we sample 50,000 points from the unit domain to supervise both occupancy and physical-field predictions. For occupancy, we adopt the semi-continuous formulation following [60]. We use a learning rate of $1e-4$, a batch size of 8 per GPU, and train for 100K steps. For the diffusion model, we employ 10 layers of DiT blocks [70], each with 16 attention heads of dimension 72. We train the diffusion model with 1000 denoising steps. For objective-guided sampling, an auxiliary U-Net surrogate predicts the task objective directly in latent code $z$. To train the diffusion model, we use a learning rate of $5e-5$, a batch size of 4 per GPU, and train for 300K. In topology-preserving refinement, we deform candidates via a 20×6×6 control-point grid along the x, y, and z axes. For the surrogate model $f_{\text{GNN}}$, we adopt MeshGraphNet [30] as our surrogate given its strong performance in mesh-based physical simulations. The surrogate is trained to predict aerodynamic drag from paired samples of geometry and ground-truth physical fields collected from the dataset. The model is trained with a learning rate of $1e-5$, a batch size of 8 per GPU, and trained for 100K. All models are trained with AdamW optimizer [71]. More training details of 3DID are included in the Appendix.

Table 1: **Quantitative comparison for aerodynamic vehicle design.** The confidence interval information is detailed in the Appendix. Note that our method shows a slight drop in coverage, mainly because the topology-preserving refinement pushes designs beyond the original distribution to achieve better aerodynamic performance.

| Method | Pred-Drag↓ | Sim-Drag↓ | Novelty↑ | Coverage↑ |
|---|---|---|---|---|
| GP, Voxel | 0.2997 | 0.4254 | 1.0399 | 0.5200 |
| GP, Voxel+PCA | 0.3059 | 0.4363 | 0.9734 | 0.5850 |
| CEM, Voxel | 0.2951 | 0.4097 | 0.9792 | 0.4350 |
| CEM, Voxel+PCA | 0.3088 | 0.4393 | 0.9864 | 0.5100 |
| CEM, TripNet | 0.3154 | 0.4161 | 1.0399 | 0.6050 |
| Backprop, Voxel | 0.2979 | 0.4146 | 0.9860 | 0.4750 |
| Backprop, Voxel+PCA | 0.3061 | 0.4614 | 0.9798 | 0.4950 |
| Backprop, TripNet | 0.3153 | 0.4170 | 1.0294 | 0.5900 |
| 3DID–NoTopoRefine (ours) | 0.2623 | 0.3766 | 0.9195 | **0.6950** |
| 3DID (ours) | **0.2607** | **0.3536** | **1.1709** | 0.4300 |

# 4 Experiments

In the experiments, we aim to answer the following questions: (1) Does 3DID outperform traditional, sampling-based, and backpropagation methods in finding high-quality designs? (2) Does our unified physics–geometry triplane representation yield better objectives than alternative latent or purely geometric embeddings? (3) Does our two-stage pipeline outperform single-stage diffusion sampling and other standard optimization methods? To answer these questions, we evaluate our method on the vehicle aerodynamic shape optimization task, a representative example of 3D inverse design.

In the following sections, we first introduce our dataset and evaluation metrics (Section 4.1). Next, we describe our experimental setup and compare against baseline methods (Section 4.2). Finally, we present two ablation studies: one on the unified physics-geometry representation (Section 4.3) and one on the two-stage optimization strategy (Section 4.4).

## 4.1 Dataset and Evaluation Details

**Dataset.** We conduct our experiments on the DrivAerNet++ dataset [72, 73], the largest available collection for aerodynamic car design, comprising over 8,000 diverse geometries paired with high-fidelity CFD simulations. For training, we use the entire dataset. We first normalize each geometry to fit within a unit cube and apply the same scaling to the simulation fields. We then uniformly sample surface point clouds with corresponding normals from the geometry. Finally, for both the occupancy field and the physical field, we adopt the data extraction strategy of Park *et al.* [74], using a grid resolution of 256. Further details of our data preparation are given in the Appendix.

**Evaluation Metrics.** We evaluate 3DID using four metrics. **Pred-Drag** is the drag coefficient estimated by a trained surrogate model, offering an approximation of the design objective. **Sim-Drag** is the drag coefficient obtained via high-fidelity CFD simulation, delivering an unbiased evaluation of aerodynamic performance. **Novelty** computes the average nearest neighbor distance from each generated design to its closest training example, indicating how distinct the designs are from existing ones. **Coverage** captures how well the generated designs cover the training distribution by measuring, for each training sample, the distance to its nearest generated design (using a k-nearest neighbor lookup) and reporting the fraction of training examples that fall within a predefined threshold. To extract features from the generated geometries, we use the pretrained PointNet model from [75]. Detailed evaluation procedures and simulation parameters are provided in the Appendix.

## 4.2 3D Vehicle Aerodynamic Design

In this experiment, we evaluate each inverse-design method on the aerodynamic shape optimization task, where the objective $\mathcal{J}$ is to reduce the drag force of the designed vehicles. All methods are trained on the same collection of car geometries paired with high-fidelity CFD simulations. As baselines, we compare against the Cross-Entropy Method(CEM) [43], the Gaussian-process surrogate with

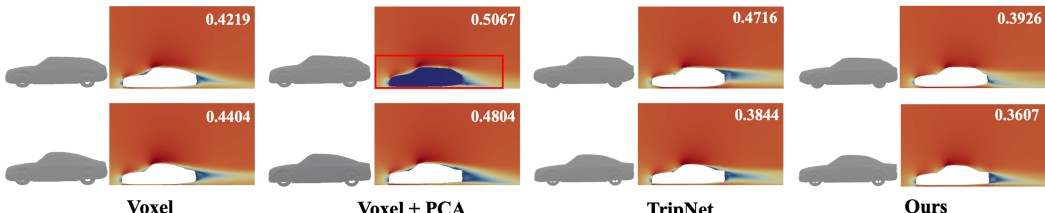

Figure 4: **Qualitative comparisons of different representations.** Each row shows four candidates with geometry (left) and simulated velocity field (right) with Sim-Drag in the top-right. Despite equal resolution, voxel methods incur higher drag and often yield non-watertight shapes (red box) due to coarse discretization. Our continuous latent representation produces watertight, smooth designs with superior aerodynamic performance, outperforming both voxel-based and geometry-only approaches.

Table 2: **Ablation study on representation choices.**

| Method | Pred-Drag↓ | Sim-Drag↓ | Novelty↑ | Coverage↑ |
|---|---|---|---|---|
| Voxel | 0.2722 | 0.4318 | 1.0683 | 0.3450 |
| Voxel+PCA | 0.2720 | 0.4565 | 0.9858 | 0.5750 |
| TripNet | 0.2698 | 0.4066 | 1.0580 | 0.5500 |
| 3DID–NoTopoRefine **(ours)** | 0.2623 | 0.3766 | 0.9195 | **0.6950** |
| 3DID **(ours)** | **0.2607** | **0.3536** | **1.1709** | 0.4300 |

Bayesian optimization(GP) [44], and the gradient-based backpropagation method(Backprop) [18]. To evaluate the impact of 3D encoding, the optimizer is instantiated with three representations: a dense voxel grid [76], a PCA-compressed voxel grid (Voxel+PCA) [22], and a pure geometry triplane network (TripNet) [21]. GP with the triplane representation is omitted due to its high computational cost. For fairness, we generate 64 candidate designs per method and report the average performance in Table 1. Architectures of baselines and training details are provided in the Appendix.

From Table 1, it can be observed that 3DID delivers the best drag force result for both pred-drag and sim-drag compared to all baselines. Specifically, our full 3DID model reduces simulated drag by 13.6% relative to the strongest baseline. These results demonstrate the effectiveness of our pipeline in discovering high-performance designs. Furthermore, our method achieves the highest novelty score (1.1709), indicating its ability to explore diverse design variations. Note that the drop in coverage occurs because topology-preserving refinement pushes designs beyond the training distribution to boost aerodynamic performance. A detailed ablation on the cascade optimization strategy is presented in Section 4.4. More visualization results and evaluation are presented in the Appendix.

### 4.3 Ablation Study on Physics–Geometry Unified Representation

In this experiment, we evaluate the performance with different representations, including Voxel [76], Voxel with PCA [22] and the pure geometry triplane (TripNet) [21]. For Voxel and Voxel with PCA, we first train a variational autoencoder (VAE) to embed the raw data into a compact latent space and then learn a diffusion model within that space. Finally, we employ our two-stage optimization pipeline consisting of gradient-guided diffusion sampling followed by topology-preserving refinement to generate diverse design candidates. Because the baseline representations do not include physical field information, we retrain a surrogate graph neural network that takes only geometry as input for the refinement stage. The results are reported in Table 2.

As shown in Table 2, our 3DID method outperforms all baselines by a wide margin in both simulation drag and novelty. Compared to the best baseline TripNet, 3DID reduces Sim Drag from 0.4066 to 0.3536, a 13.0% improvement, and lowers Pred Drag by 3.4% (from 0.2698 to 0.2607). It also increases novelty from 1.0683 to 1.1709, a 9.6% gain. In Figure 5, our continuous latent representation consistently yields watertight smooth geometries with superior aerodynamic performance, whereas voxel-based methods suffer higher drag and non-watertight artifacts. Although TripNet embeds continuous geometry, its optimized designs remain inferior. We attribute this to the absence of physical field guidance, which weakens the optimization gradients in the refinement stage.

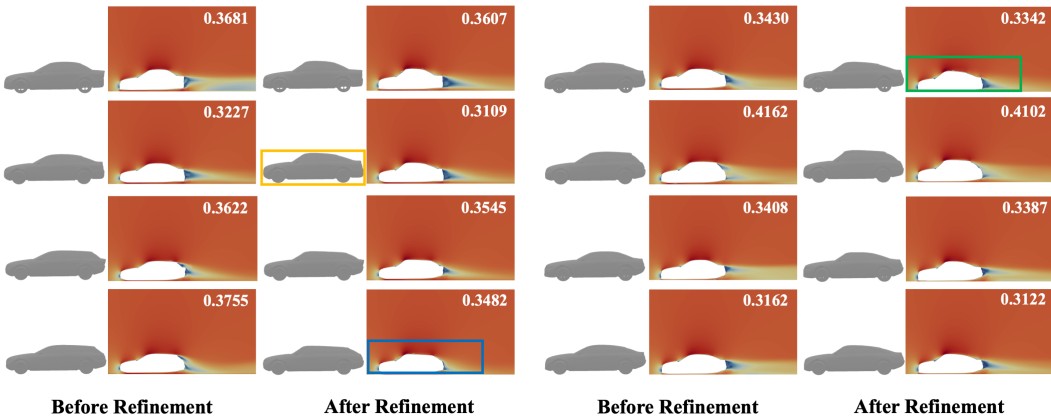

Figure 5: **Qualitative comparisons of topology-preserving refinement.** Each row presents two design candidates comparisons with their geometry and simulated velocity field heatmaps. Sim-Drag values are shown in the top-right corner of each panel. Refined designs exhibit a more significant fastback profile (yellow box), reduced low-velocity recirculation zones (blue box), and stronger downward flow (green box), indicating improved aerodynamic performance.

Table 3: **Ablation study on design strategies.**

| Method | Pred-Drag↓ | Sim-Drag↓ | Novelty↑ | Coverage↑ |
|---|---|---|---|---|
| CEM | 0.3152 | 0.3987 | 1.0730 | 0.6800 |
| GD | 0.3023 | 0.4095 | 1.0878 | 0.5800 |
| W/O Guidance | 0.2971 | 0.3944 | 0.9177 | **0.7104** |
| 3DID–NoTopoRefine **(ours)** | 0.2623 | 0.3766 | 0.9195 | 0.6950 |
| 3DID **(ours)** | **0.2607** | **0.3536** | **1.1709** | 0.4300 |

## 4.4 Ablation Study on Two-Stage Optimization Strategy

In this experiment, we validate the effectiveness of our two-stage optimization pipeline by comparing it against two alternative design strategies, including Cross-Entropy Method (CEM) and gradient descent (GD), as well as a diffusion-only sampling approach without objective gradient guidance. All methods are based on our physics-geometry unified representation.

From Table 3, we see that our full 3DID outperforms all baselines in Pred-Drag, Sim-Drag, and Novelty. Notably, when designing without guidance, our diffusion model attains the highest coverage value of 0.7104, as it captures the data manifold comprehensively. When generating without guidance, the diffusion model tends to mimic the distribution of the dataset, which leads to an increase of coverage but lacks the targeted optimization for aerodynamic performance and novelty that our 3DID method provides. Furthermore, to better validate the effectiveness of our topology-preserving refinement stage, we provide side-by-side qualitative comparisons in Figure 5. It can be seen that after the refinement stage, each candidate develops a more significant fastback profile, with diminished low-velocity recirculation regions, and stronger downward flow patterns, all indicators of improved aerodynamic performance as confirmed by the reduced Sim-Drag values.

## 5 Limitations

**Limited to static physical fields.** Despite the fact that 3DID achieves impressive results, a significant limitation is its focus on static fields. The current framework does not support inverse design involving time-dependent or dynamic physical fields. Time-dependent physical systems often involve solid geometries coupled with evolving physical properties over time. This would pose challenges for representation and optimization within our framework. Enhancing 3DID with time-aware representations and models may address these limitations, which we leave as an important direction for future work.

**Limited to single objective optimization.** In 3DID, we address the inverse problem with a single objective, which may limit its applicability for broader scenarios. Although it is straightforward to aggregate multiple objectives into a single composite loss, this approach may overlook potential conflicts and trade-offs between objectives. Extending 3DID to support true multi-objective optimization is a promising direction for future research.

**Limited to surrogate-based physics awareness.** We incorporate physical fields via data-driven surrogates and joint geometry–physics embeddings during generation and refinement, rather than enforcing governing laws explicitly. This guides designs toward plausible, high-performing regions but does not enforce hard physical constraints. Exploring hard-constraint mechanisms or tighter PDE-consistent couplings is an important direction for future work.

# 6   Conclusion

In this paper, we tackle the problem of 3D inverse design, which faces challenges from the high-dimensional physics-geometry coupling and the exploration–validity trade-off. To represent the coupled space, we propose a physics-geometry unified representation that preserves fine-grained shape details and physical-field information while significantly reducing dimensionality. Based on this representation, we introduce a two-stage physics-aware optimization strategy that first explores the latent manifold via gradient-guided diffusion sampling and then refines candidates through topology-preserving refinement. Extensive experiments demonstrate that our 3DID framework generates high-fidelity 3D models with greater versatility and superior performance on target objectives.

# Acknowledgments

This work is partially supported by National Science and Technology Major Project (2022ZD0117802). This work was also supported in part by "Pioneer" and "Leading Goose" R&D Program of Zhejiang (No. 2025C02032), the Fundamental Research Funds for the Central Universities (226-2025-00080), and the Earth System Big Data Platform of the School of Earth Sciences, Zhejiang University.

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

# Appendix

In Appendix A, we provide additional experiments results.

- In Appendix A.1, we further provide the full results for 3D vehicle aerodynamic design.
- In Appendix A.2, we provide more visualization of design results.
- In Appendix A.3, we provide more qualitative comparisons of topology-preserving refinement.

Appendix B: The implementation details of baseline methods.

Appendix C: The dataset processing details.

Appendix D: The implementation details of 3DID.

Appendix E: The evaluation details of 3DID.

Appendix F: The broader impact of 3DID.

Appendix G: The licenses of datasets, codes, and models used in this paper.

# A  Additional Results

## A.1  Full Results for 3D Vehicle Aerodynamic Design

Here we present the full statistical results of our experiments, including 95% confidence intervals for all compared methods, shown in Table 4. A box plot of the simulation-derived drag coefficient (Sim-Drag) is shown in Figure 6, illustrating the distribution, variability, and outlier behavior across different approaches.

Table 4: **Quantitative comparison for aerodynamic vehicle design.**

| Method | Pred-Drag↓ | Sim-Drag↓ | Novelty↑ | Coverage↑ |
|---|---|---|---|---|
| GP, Voxel | 0.2997±0.0436 | 0.4254 ± 0.0351 | 1.0399±0.0572 | 0.5200±0.0675 |
| GP, Voxel+PCA | 0.3059±0.0490 | 0.4363 ± 0.0425 | 0.9734±0.0195 | 0.5850±0.0675 |
| CEM, Voxel | 0.2951±0.0421 | 0.4097 ± 0.0279 | 0.9792±0.0213 | 0.4350±0.0676 |
| CEM, Voxel+PCA | 0.3088±0.0478 | 0.4393 ± 0.0469 | 0.9864±0.0250 | 0.5100±0.0600 |
| CEM, TripNet | 0.3154±0.0476 | 0.4161 ± 0.0415 | 1.0399±0.0323 | 0.6050± 0.0725 |
| Backprop, Voxel | 0.2979±0.0314 | 0.4146 ± 0.0244 | 0.9860±0.0204 | 0.4750±0.0675 |
| Backprop, Voxel+PCA | 0.3061±0.0576 | 0.4614 ± 0.0316 | 0.9798±0.0208 | 0.4950±0.0675 |
| Backprop, TripNet | 0.3153±0.0472 | 0.4170 ± 0.0444 | 1.0294±0.0290 | 0.5900±0.0700 |
| 3DID–NoTopoRefine (**ours**) | 0.2623±0.0373 | 0.3766 ± 0.0393 | 0.9195±0.0213 | **0.6950**±0.0627 |
| 3DID (**ours**) | **0.2607**±0.0331 | **0.3536**± 0.0313 | **1.1709**±0.0282 | 0.4300±0.0650 |

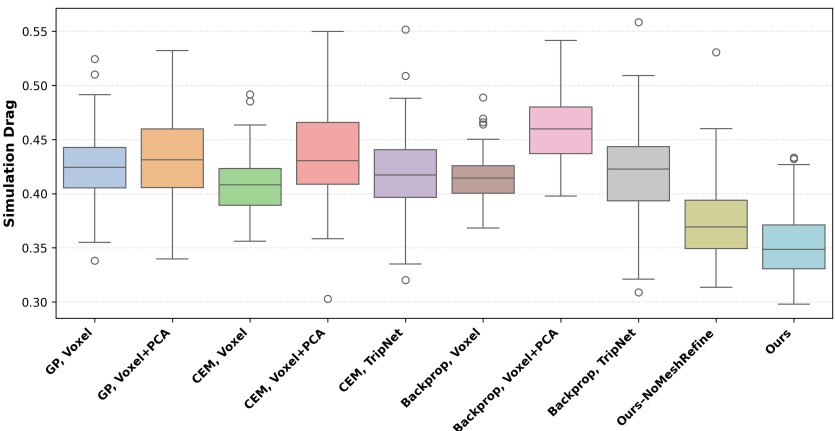

Figure 6: **The box plot of the simulation-derived drag coefficient.**

## A.2  Visualization of 3DID Design

Additional visualizations of our designs are provided in Figure 7, where each design is shown alongside its geometry and corresponding physical fields.

## A.3  Comparisons of Topology-Preserving Refinement

We provide additional qualitative comparisons in Figure 8 to demonstrate the effectiveness of our refinement stage. As shown, the design candidates consistently evolve toward a fastback profile after refinement, exhibiting reduced low-velocity recirculation regions and enhanced downward flow patterns, which indicate improved aerodynamic performance.

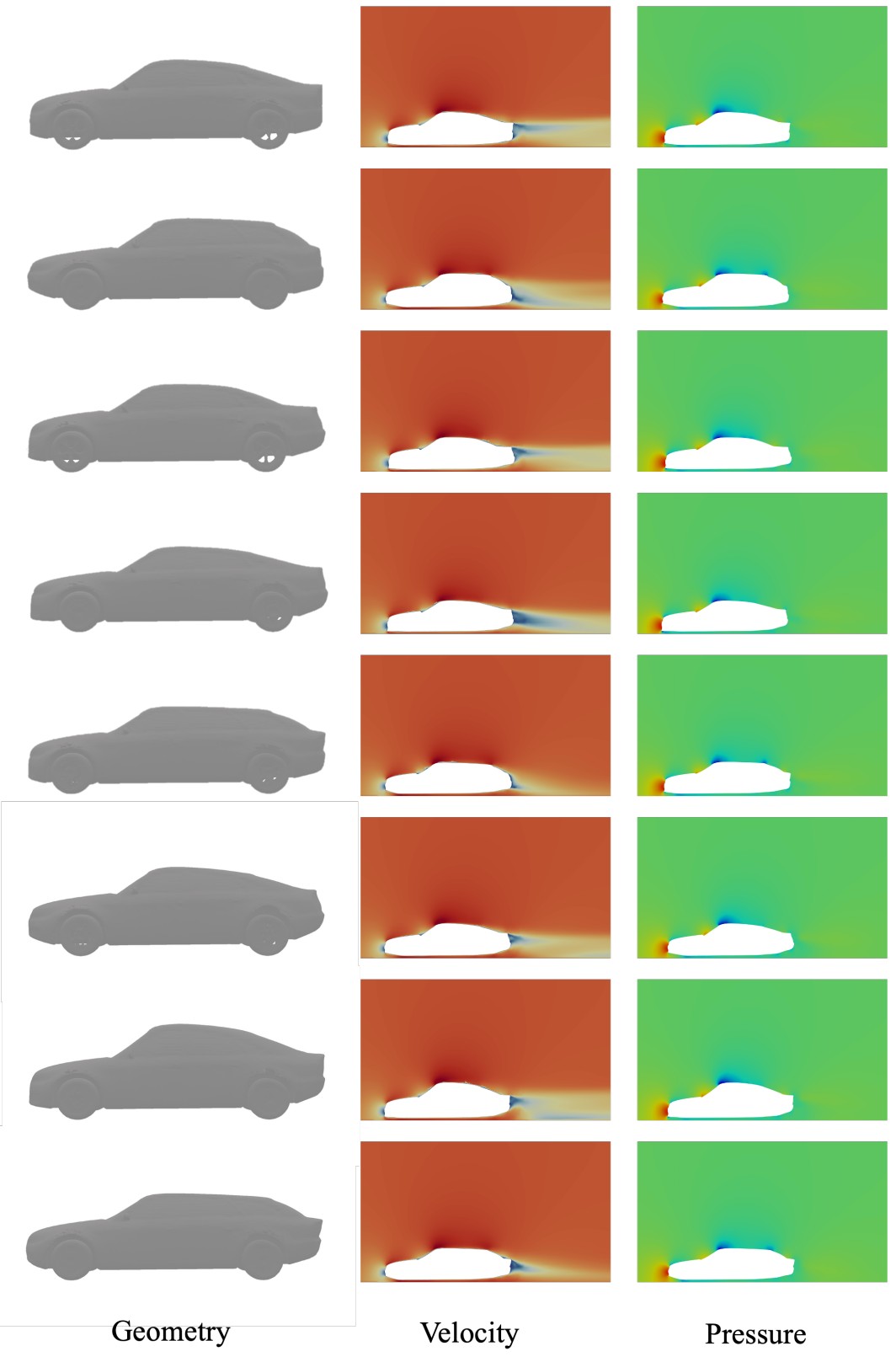

Geometry          Velocity          Pressure

Figure 7: **Qualitative results of our 3DID.** Each row displays a design candidate along with its corresponding velocity and pressure field heatmaps.

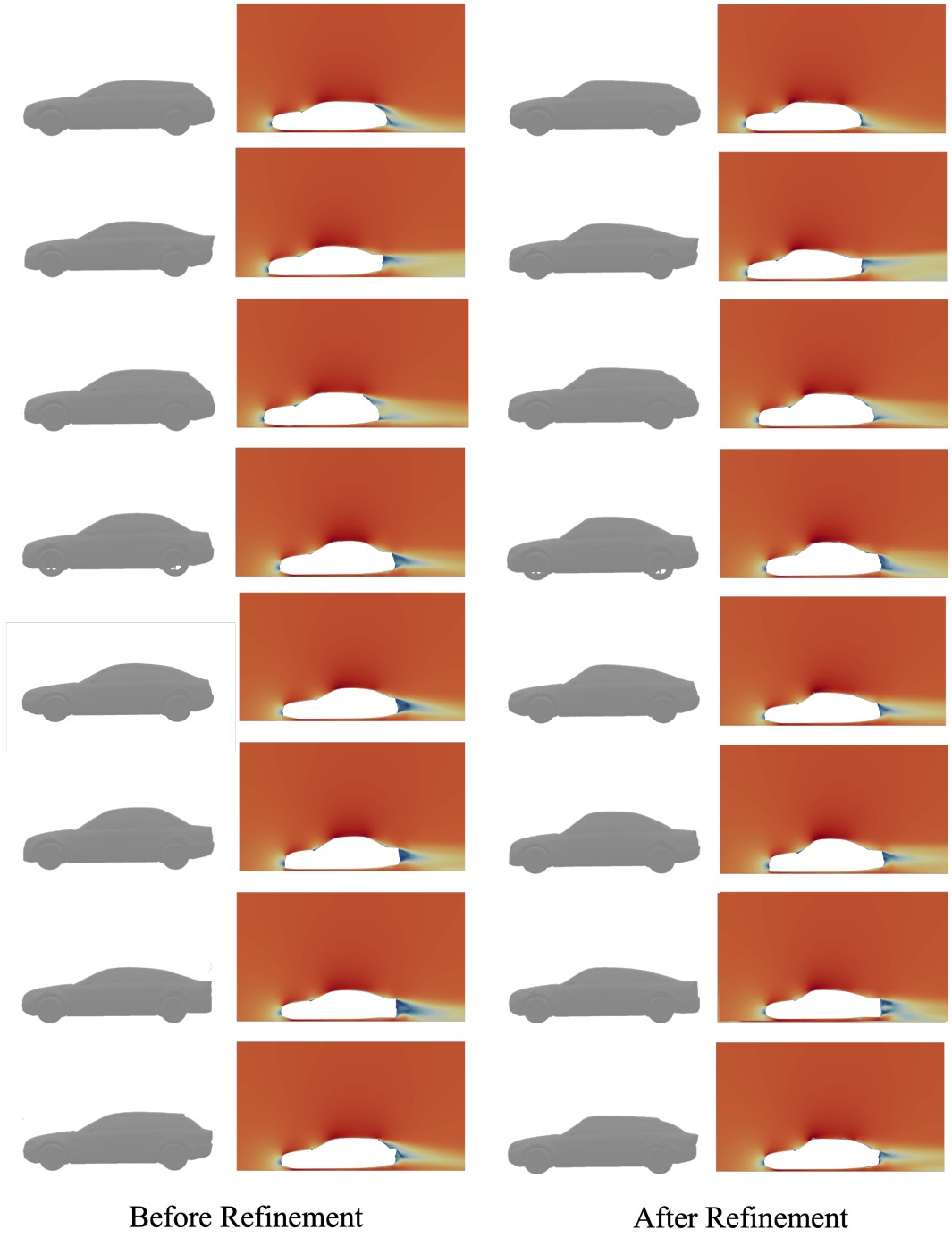

Before Refinement          After Refinement

Figure 8: **Qualitative comparisons of topology-preserving refinement.** Each row presents two design candidates comparisons with their geometry and simulated velocity field heatmaps.

# B  Baseline implementation details

In our experiments, we compare our method against traditional sampling-based and backpropagation-based approaches using various design representations. As baselines, we include the Cross-Entropy Method(CEM) [43], the Gaussian-process surrogate with Bayesian optimization(GP) [44], and the gradient-based backpropagation method(Backprop) [18]. For representations, the optimizer is instantiated with three representations: a dense voxel grid [76], a PCA-compressed voxel grid (Voxel+PCA) [22], and a pure geometry triplane network (TripNet) [21]. For each representation, we train a VAE model [77] to compress the high-dimensional geometry into a compact latent code, which serves as the optimization space for inverse design.

## B.1  Representation Baseline

**Voxel.** We train a voxel VAE [77] model directly on dense voxelized geometry to learn a latent embedding, as demonstrated in Figure 9. To train the model, we utilize the entire DrivAerNet++ [72] dataset, and voxelize the provided geometry with $256^3$ resolution. For the Encoder, we leverage a sequence of 3D convolution layers followed by batch normalization and LeakyReLU to encode the voxel grid into a compact latent $z_{\text{voxel}}$. For voxel decoder, the latent vector $z_{\text{voxel}}$ is first projected to a high-dimensional feature space and reshaped into a 3D tensor. A sequence of 3D transposed convolutional layers is then applied to reconstruct the voxel grid from this intermediate representation. Additionally, a separate drag prediction head, implemented as a multi-layer perceptron (MLP), is applied to estimate the target drag coefficient. We train the VAE model with reconstruction loss $\mathcal{L}_{\text{recon}}$, KL loss $\mathcal{L}_{\text{KL}}$, and the drag coefficient prediction loss $\mathcal{L}_{\text{drag}}$. The hyperparameters of the model and training are provided in Table 5

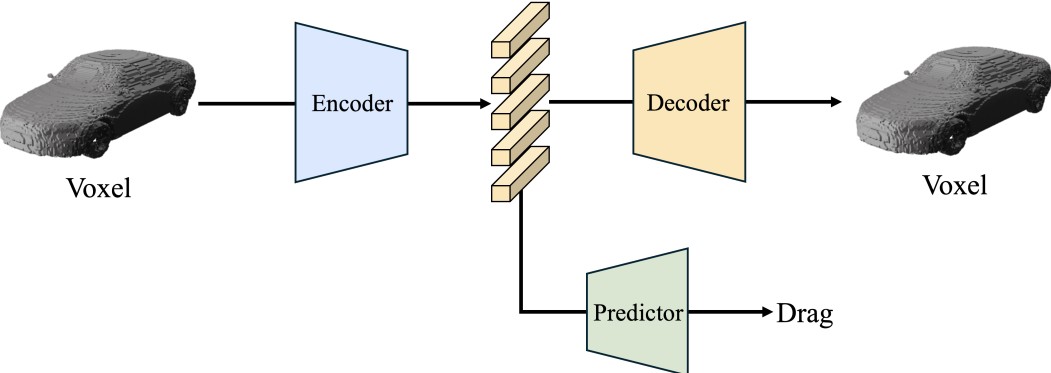

Figure 9: **The overview of Voxel-VAE.**

**Voxel-PCA.** Our Voxel-PCA representation is based on the representation proposed by [22] with modifications. In contrast to the Voxel-VAE, which directly uses voxel grids as input, the Voxel-PCA model first applies a dimensionality reduction step before downstream processing, as shown in Figure 10. Specifically, given the voxel data, we perform PCA [78] to obtain a compact representation of each geometry. Then, with this representation, we leverage a series of MLPs to encode the reduced features into a latent code $z_{\text{voxel-pca}}$. For reconstruction, an MLP decoder is first applied to reconstruct the PCA features from the latent code, which are then projected back to the voxel grid using the inverse PCA transformation. For drag prediction, similar to Voxel-VAE, a separate drag prediction head is applied to estimate the target drag coefficient. Our Voxel-PCA model is also trained with reconstruction loss $\mathcal{L}_{\text{recon}}$, KL loss $\mathcal{L}_{\text{KL}}$ and drag prediction loss $\mathcal{L}_{\text{drag}}$. The hyperparameters of the model and training are provided in Table 6.

**TripNet.** Our TripNet representation is a pure geometry-based triplane representation, similar to the one proposed in [21], where it was used for forward prediction. The training procedure mirrors that of our unified physics-geometry framework, but excludes the physical field prediction branch, as illustrated in Figure 11. To obtain the representation, we utilize transformers with learnable tokens to extract features from the input point cloud. These features are then decoded using a transformer-based decoder and a geometry mapping network to predict the occupancy field of the design geometry. We

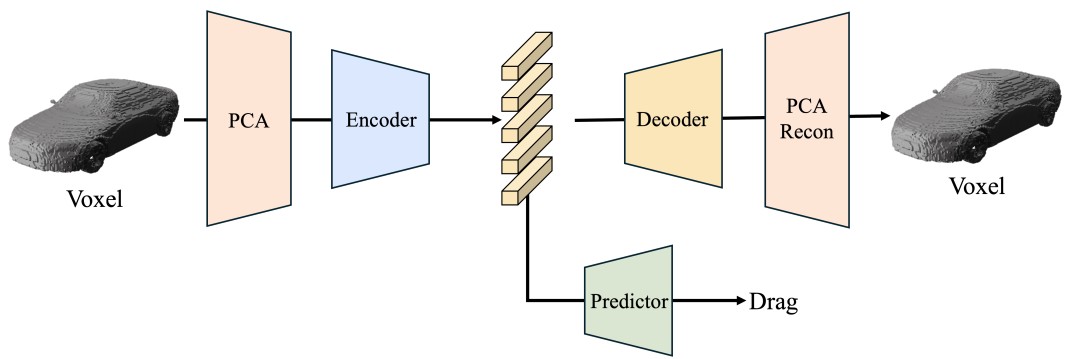

Figure 10: **The overview of Voxel-PCA-VAE.**

utilize the Binary Cross-Entropy loss $\mathcal{L}_{\text{BCE}}$ and KL loss $\mathcal{L}_{\text{KL}}$ to supervise the training of VAE. To further predict the drag coefficient, we adopt the same U-Net architecture used in our objective-guided diffusion model. The TripNet-VAE architecture adopts the same hyperparameter configuration as the geometry branch of our PG-VAE. More training hyperparameters are provided in Table 7.

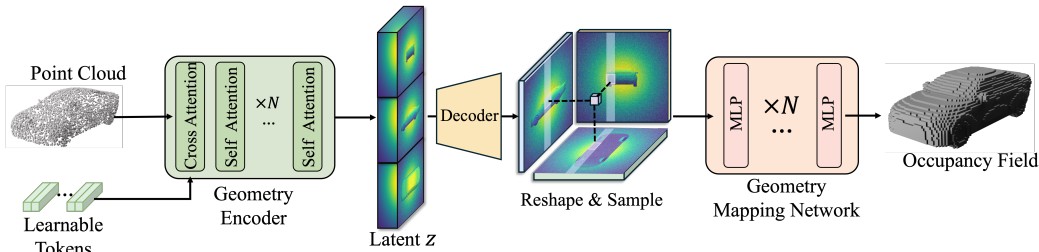

Figure 11: **The overview of TripNet-VAE.**

## B.2 Optimization Baseline

**CEM.** Cross Entropy Method [43] is a traditional sampling-based optimization method widely used in classical inverse design problems. It starts with an initial distribution, and in each iteration, it samples multiple candidates from the current distribution. Then, these candidates are evaluated against the target objective function to select a subset of elite samples with the best performance. The distribution parameters are updated based on these elite samples. A smoothing coefficient controls the rate of distribution updates between iterations. This process continues until convergence or a maximum number of iterations is reached. In our experiment, we utilize a Gaussian distribution derived from encoded randomly selected samples as the initial distribution to provide a valid starting point.

**GP.** Gaussian-process surrogate with Bayesian optimization is a classical optimization method for black-box optimization [16, 17]. Bayesian optimization (BO) operates by constructing a probabilistic surrogate model, commonly a Gaussian-process model, to approximate the objective function based on past observations. At each iteration, an acquisition function is used to balance exploration of uncertain regions and exploitation of promising areas, guiding the selection of the next evaluation point. This strategy enables efficient optimization in high-cost or sample-limited scenarios by focusing evaluations on the most informative regions of the design space. While this method is effective in low-dimensional settings, constructing an accurate GP model becomes computationally expensive and challenging as the dimensionality of the design space increases. Therefore, GP-based Bayesian optimization is typically limited to small-scale or low-dimensional problems, where the surrogate can be reliably trained. In our experiment, our Gaussian process employs a Matérn kernel with constant and white noise components to model the objective function. At each iteration, Expected Improvement (EI) is used as the acquisition function.

**Backprop.** With the trained surrogate models, end-to-end backpropagation enables efficient gradient-based optimization of the design, leveraging the differentiability of the surrogate to guide updates [18, 19]. In our experiments, we use the trained drag predictor as the surrogate and update the latent code using the Adam optimizer.

## C    Dataset processing details.

In this work, we conduct experiments on DrivAerNet++ [72], which is the largest aerodynamic car design dataset, comprising diverse car designs with corresponding CFD simulations. To train our model, we use the dataset with 8085 car designs to extract the point cloud and physical field. We first normalize each geometry of cars to fit within a unit cube, then uniformly sample 50,000 points with corresponding normals from the geometry surface. For the physical field, we apply the same scaling factor to ensure alignment with the normalized geometry. Subsequently, we randomly sample 50,000 points within the unit cube and interpolate the physical field values at each location. These points serve as the input of our PG-VAE. For supervision, we additionally sample another 50,000 points, each annotated with both occupancy values and physical field data. In this work, we focus on the pressure and velocity fields for the physical field representation, as wall shear stress is defined only on the surface of the geometry and is thus not suitable for volumetric sampling. During physical field interpolation, since some DrivAerNet++ samples are simulated using only half of the geometry, we map each sampled point to its symmetric counterpart when necessary. For the U-Net and GNN surrogate models used in guided diffusion sampling and topology-preserving optimization, we employ the drag coefficient values provided by the DrivAerNet++ dataset as ground truth supervision during training.

## D    Implementation details.

Our framework consists of three key components: the Physics–Geometry VAE (PG-VAE), Objective-Guided Diffusion, and Topology-Preserving Refinement. Below, we provide detailed implementation descriptions for each component.

**PG-VAE.** The PG-VAE serves to compress both the design geometry and the corresponding physical field into a unified latent representation. We sample $N_g = N_p = 50{,}000$ points for the geometry and physical field branches, respectively. The encoder consists of one cross-attention layer and eight self-attention layers, each with 12 attention heads and an embedding dimension of $d_z = 64$. We use $r = 64$ for learnable tokens, and each with a channel dimension of $d_e = 768$, to enhance representation expressiveness. The latent code dimension is set to $d_z = 32$. The decoder architecture consists of one self-attention layer followed by five ResNet blocks [69], which upsample the latent vector into a triplane representation with resolution $R = 256$ and channel dimension $d_t = 64$. The output triplane is then queried using a mapping network composed of five fully connected layers with a hidden size of 32 per branch. We adopt a semi-continuous occupancy formulation [60] and supervise both occupancy and physical field predictions using 50,000 sampled points within the normalized unit cube. We optimize the VAE using a combination of three loss terms: binary cross-entropy loss ($\lambda_{\mathrm{BCE}} = 10^{-3}$), mean squared error for field regression ($\lambda_{\mathrm{MSE}} = 10^{-5}$), and KL divergence ($\lambda_{\mathrm{KL}} = 10^{-6}$). Training is performed using the AdamW optimizer [71] with a learning rate of $1 \times 10^{-4}$, batch size 8 per GPU, for 100,000 steps. We use four NVIDIA RTX A6000 GPUs to train the model.

**Objective-Guided Diffusion.** To explore the latent design space efficiently, we employ a latent-space diffusion model composed of 10 DiT blocks [70], each containing 16 attention heads with a head dimension of 72. The diffusion process includes 1,000 denoising steps. During inference, an auxiliary U-Net surrogate network is used to predict the task objective directly from the latent code $z$, thereby guiding the sampling process toward optimal designs. The diffusion model is trained using a learning rate of $5 \times 10^{-5}$, batch size of 4 per GPU, for 300,000 steps with the AdamW optimizer. We use four NVIDIA RTX A6000 GPUs to train the diffusion model.

**Topology-Preserving Refinement.** To refine the initial design candidates while maintaining mesh topology, we apply a Free-Form Deformation (FFD) grid with $20 \times 6 \times 6$ control points along the x, y, and z axes, respectively. The deformation is guided by a surrogate model based on Mesh-GraphNet [30], which comprises 8 message-passing blocks and operates on the surface mesh. The

MeshGraphNet is trained to predict the drag force from a deformed mesh, serving as a differentiable objective function during refinement. This model is trained with a learning rate of $1 \times 10^{-5}$, batch size of 8 per GPU, for 100,000 steps, using AdamW as the optimizer. We use two NVIDIA RTX A6000 GPUs to train the MeshGraphNet.

## E Evaluation details.

In our experiments, we evaluate the design candidates using four metrics: predicted drag force (Pred-Drag), simulated drag force (Sim-Drag), novelty, and coverage.

**Pred-Drag.** We use the pretrained surrogate model to estimate the drag force of each candidate mesh. Given the mesh of designed candidates $M^*$, our surrogate model directly predict the objective drag force $\hat{\mathcal{J}}$ which can be formalized as:

$$\hat{\mathcal{J}} = \mathcal{F}_{\text{surrogate}}(M^*), \tag{12}$$

where $\mathcal{F}_{\text{surrogate}}$ denotes the learned mapping from 3D mesh geometry to the predicted drag coefficient. For our surrogate model, we adopt a MeshGraphNet [30] with 8 message passing blocks as the surrogate model. Unlike the model used in our topology-preserving refinement stage, this predictor operates solely on geometry, without requiring the associated physical field. To train the model, we use the entire DrivAerNet++ [72] dataset. Given that different representations may produce varying topological structures, we apply remeshing and simplification to all candidates for fair comparison.

**Sim-Drag.** To obtain an unbiased evaluation of the generated designs, we perform high-fidelity Computational Fluid Dynamics (CFD) simulations and compute the corresponding drag coefficients. Following DrivAerNet++ [72], we employ the OpenFOAM®V11 [79] to conduct steady-state incompressible simulation using the $k - \omega$ SST turbulence model, based on Menter's formulation [80]. We performed a series of quality checks to ensure the generated geometries were simulation-ready and properly aligned within the CFD domain. During simulation, considering the computation cost, we set the maximum local cells to 10 million and the maximum global cells to 50 million in snappyHexMesh. The simulation iterates for 1000s, and we use the final 30% simulation data to calculate the average drag coefficient. The hyperparameters of our simulation are provided in Table 8.

**Novelty.** To quantitatively assess how different the generated designs are from the training data, we measure the novelty of each candidate. Specifically, novelty is computed as the average distance from each generated design to its nearest neighbor in the training set, reflecting how distinct the generated designs are from existing ones. Let $\{g_i\}_{i=1}^{N_g}$ denote the set of generated designs and $\{t_j\}_{j=1}^{N_t}$ denote the set of training designs in the feature space. The novelty is defined as:

$$\text{Novelty} = \frac{1}{N_g} \sum_{i=1}^{N_g} \min_j d(g_i, t_j), \tag{13}$$

where $d(\cdot, \cdot)$ denotes the distance between feature embeddings, computed using the pretrained PointNet encoder [75].

**Coverage.** The coverage metric (also known as recall) evaluates how well the generated designs cover the training distribution by measuring, for each training sample, the distance to its nearest generated design (using a k-nearest neighbor lookup) and reporting the fraction of training examples that fall within a predefined threshold. Let $\{g_i\}_{i=1}^{N_g}$ denote the set of generated designs and $\{t_j\}_{j=1}^{N_t}$ denote the set of training designs in the feature space. The coverage is defined as:

$$\text{Coverage} = \frac{1}{N_t} \sum_{j=1}^{N_t} \mathbf{1}[\min_i d(t_j, g_i) \leq \tau] \tag{14}$$

where $d(\cdot, \cdot)$ is a distance metric, $\tau$ is a predefined threshold, and $\mathbf{1}[\cdot]$ is the indicator function that equals 1 if the condition is true and 0 otherwise.

# F  Broader Impacts

**Academic Impact.** 3DID's methodology, which enables direct navigation through 3D physics-geometry space, simplifies the 3D inverse design process. With the unified physics-geometry representation, the computation gap between 3D and lower-dimensional inverse design is narrowed, allowing researchers to focus more on exploring cutting-edge inverse design strategies rather than being constrained by computational limitations. With the two-stage optimization strategy, our method balances between exploration and validity, offering researchers an effective approach for inverse design involving 3D geometry.

**Social Impact.** The proposed 3D Inverse Design (3DID) framework extends the scope of geometry-driven design by enabling direct optimization of full 3D structures from scratch. By combining unified physics-geometry representations with physics-aware optimization, our method opens the door to more efficient, automated design workflows in fields such as aerospace engineering, biomedicine, additive manufacturing, and nanophotonics. In particular, 3DID can be applied to complex design tasks that traditionally rely on expert-crafted initial geometries and time-consuming simulation-based evaluations. In mechanical engineering, it can be used to optimize structural components for strength, weight, and thermal performance without manual trial-and-error. In the medical field, 3DID enables the fabrication of patient-specific implants by automatically generating geometries tailored to individual physiological and functional requirements.

# G License

The code will be publicly accessible. We use standard licenses from the community. We include the following licenses for the codes, datasets, and models we used in this paper.

1. **Dataset**
   - DrivAerNet++ [72]: CC BY-NC 4.0
2. **Codes**
   - NVIDIA PhysicsNeMo: Apache License 2.0
3. **Evaluation**
   - OpenFOAM [79]: GNU General Public License

Table 5: **Hyperparameters for Voxel-VAE**

| Hyperparameter name | Value |
|---|---|
| **Hyperparameters for Voxel-VAE architecture:** | |
| Input shape | [8, 256, 256, 256] |
| Output shape | [8, 256, 256, 256] |
| Number of 3D convolution layer | 5 |
| Dimension of latent $z_{\text{voxel}}$ | 512 |
| Number of 3D transposed convolutional layer | 5 |
| Number of MLPs in drag predictor | 5 |
| Batch size | 8 |
| Dimension of encoder | (1, 32, 64, 128, 256, 512) |
| Dimension of voxel decoder | (512, 256, 128, 64, 32, 1) |
| Dimension of drag predictor | (512, 256, 128, 64, 32, 1) |
| **Hyperparameters for Voxel-VAE training:** | |
| Optimizer | AdamW |
| Learning rate | $1e-4$ |
| Learning steps | 100K |
| Learning rate adjustment strategy | Cosine |
| Warm-up steps | 5K |
| $\mathcal{L}_{\text{recon}}$ weight | $10^{-3}$ |
| $\mathcal{L}_{\text{KL}}$ weight | $10^{-4}$ |
| $\mathcal{L}_{\text{drag}}$ weight | $10^{-3}$ |

Table 6: **Hyperparameters for Voxel-PCA-VAE**

| Hyperparameter name | Value |
| --- | --- |
| **Hyperparameters for Voxel-PCA-VAE architecture:** | |
| PCA output dimension | 400 |
| Number of MLP layers in encoder | 4 |
| Dimension of latent $z_{\text{voxel-pca}}$ | 64 |
| Number of MLP layers in decoder | 4 |
| Number of MLPs in drag predictor | 2 |
| Batch size | 32 |
| Dimension of encoder | (400, 256, 128, 64, 64) |
| Dimension of PCA decoder | (64, 64, 128, 256, 400) |
| Dimension of drag predictor | (64, 32, 1) |
| **Hyperparameters for Voxel-PCA-VAE training:** | |
| Optimizer | AdamW |
| Learning rate | $5e-4$ |
| Learning steps | 100K |
| Learning rate adjustment strategy | Cosine |
| Warm-up steps | 5K |
| $\mathcal{L}_{\text{recon}}$ weight | $10^{-2}$ |
| $\mathcal{L}_{\text{KL}}$ weight | $10^{-4}$ |
| $\mathcal{L}_{\text{drag}}$ weight | $10^{-3}$ |

Table 7: **Hyperparameters for TripNet VAE**

| Hyperparameter name | Value |
| --- | --- |
| **Hyperparameters for TripNet-VAE training:** | |
| Batch size | 8 |
| Optimizer | AdamW |
| Learning rate | $1e-4$ |
| Learning steps | 100K |
| Learning rate adjustment strategy | Cosine |
| Warm-up steps | 5K |
| $\mathcal{L}_{\text{BCE}}$ weight | $10^{-3}$ |
| $\mathcal{L}_{\text{KL}}$ weight | $10^{-6}$ |

Table 8: **CFD Simulation Parameters for OpenFOAM**

| Parameter name | Value |
|---|---|
| **Solver Configuration:** | |
| OpenFOAM version | v11 |
| Solver | incompressibleFluid |
| Algorithm | SIMPLE |
| Turbulence model | k-$\omega$-SST |
| Simulation type | Steady-state RANS |
| **Flow Conditions:** | |
| Flow velocity ($u_\infty$) | 30 m/s |
| Kinematic viscosity ($\nu$) | $1.56 \times 10^{-5}$ m²/s |
| Air density ($\rho$) | 1.184 kg/m³ |
| Turbulent kinetic energy (k) | 0.375 m²/s² |
| Specific dissipation rate ($\omega$) | 1.78 s$^{-1}$ |
| **Computational Domain:** | |
| Domain dimensions | 44×8×6.4 m |
| Inlet distance | 12 m upstream |
| Outlet distance | 32 m downstream |
| **Solver Tolerances:** | |
| Pressure absolute tolerance | $1 \times 10^{-6}$ |
| Pressure relative tolerance | $3 \times 10^{-2}$ |
| Velocity absolute tolerance | $1 \times 10^{-8}$ |
| Velocity relative tolerance | $5 \times 10^{-3}$ |
| Turbulence absolute tolerance | $1 \times 10^{-8}$ |
| Turbulence relative tolerance | $1 \times 10^{-3}$ |
| Potential solver absolute tolerance | $1 \times 10^{-7}$ |
| Potential solver relative tolerance | $1 \times 10^{-2}$ |
| **Mesh Refinement:** | |
| Surface refinement level | 3-4 |
| Feature refinement level | 4 |
| Regional refinement level | 2 |
| Wake refinement level | 2 |
| Boundary layers | 5 layers |
| Layer expansion ratio | 1.2 |
| Final layer thickness | 0.5 |
| **Force Calculation:** | |
| Reference length ($l_{ref}$) | 4.777 m |
| Reference area ($A_{ref}$) | 2.0 m² |
| Reference center | (0, 0, 0) |
| Drag direction | (1, 0, 0) |
| Lift direction | (0, 0, 1) |
| **Simulation Control:** | |
| End time | 1000 s |
| Time step | 1 s |
| Write interval | 100 steps |
| Force coeffs write interval | 10 steps |

