# OpenReview forum: "3DID: Direct 3D Inverse Design for Aerodynamics with Physics-Aware Optimization"
_NeurIPS.cc/2025/Conference — NeurIPS 2025 poster_

### Official Review · Reviewer_ht5R · 2025-06-19

**Clarity:** 3
**Significance:** 2
**Originality:** 2
**Rating:** 4
**Confidence:** 3

**Summary:**

This paper performs inverse design of car geometries for optimizing aerodynamics properties, e.g., drag. The paper creates a latent space that encodes both the geometry of the cars and the simulated physical fields around the car geometry. The paper then uses this latent space for optimizing for a new shape (in the latent space). As a final post processing step, the paper further optimizes the obtained design through a shape optimization that preserves the topology. The paper compares its method to two other inverse design method and reports superiority. It also studies some of its own design decision through ablations.

**Questions:**

- On the amortization note above, how general the surrogation is? With the 8000 training data, can one use the trained surrogates for slightly changed conditions/applications?

- How much topology variation exist in the training data and how much the topology of optimized designs is different from the training data? Are we talking about mesh-topology or the more macro-scale car topology?

- I had difficulty understanding 2D projection part. Some clarification would be helpful to the paper.

- Equation 1 is hard to parse mathematically. Please check the correctness.

**Ethical Concerns:**

["NO or VERY MINOR ethics concerns only"]

**Final Justification:**

I think this is a good paper and the rebuttal helped with some clarifications. I am still not sure how my main concern will be reflected in the paper which I believe is quite fundamental:
"The authors performed a suggested experiment where they check whether any generated design (using their method) can beat the best performing design in the training set. And the answer is no (I command them for their transparency). As I mentioned in the review, this is a sanity check. This means that the 'interpolated' designs cannot achieve better performance. In other words, the method is not capable of giving us better designs but only different designs."

**Limitations:**

Not much discussed in the main paper. Some of the points raised in Strengths/Weakness section above could potentially be used.

**Paper Formatting Concerns:**

None.

**Quality:**

2

**Strengths And Weaknesses:**

Strengths:
- The choice of the problem: this problem is inherently difficult and underrepresented in ML.

- Validation is relatively good and the paper has done the right thing by including how a ground-truth (and not only surrogate) rates the optimized design, through 'Sim-Drag' metric.

Weaknesses:
- The two main contributions of the paper could have been more discussed. For example, there should have been references to the works that encode both design and physical properties as well as works that do refinement.

- An important missing evaluation that acts as a sanity check is to compare the performance of the optimized design with the best performing design in the training dataset.

- The whole point of surrogation is to amortize its training during inference. Is this the case in this paper? The paper shows the best performing shape through surrogate but what if we optimize for 1000 different drags and show the statistics? Ideally, this must be coupled with other objectives and have a multi-objective optimization.

- I am not sure why this method is called a physics-aware optimization. The fact that the physical fields are used jointly for training the encoder, does not fit the (loose) definition of physics-aware.

- Minor, but the title is a bit too broad by not including the context (car geometries and aerodynamics).

---

> ### Author Rebuttal · Authors · 2025-07-31
>
> We thank the reviewer for the detailed feedback. We appreciate the recognition of our effort in validating results with simulated drag and in tackling a difficult but underrepresented problem. Below, we address each comment point-by-point.
>
> > **Comment 1: "there should have been references to the works that encode both design and physical properties as well as works that do refinement."**
>
> **A1:** We thank the reviewer for pointing this out. We will incorporate additional references to prior works in the revised version.
>
> > **Comment 2: "An important missing evaluation that acts as a sanity check is to compare the performance of the optimized design with the best performing design in the training dataset."**
>
> **A2:** We conducted a comparison between the best-performing designs in the training set and our generated samples. Among the 100 generated designs, none directly outperformed the best-performing design from the training dataset, as shown in Table 1. This occurs because our diffusion model aims to generate diverse geometries across the entire data distribution rather than specifically targeting the global optimum.
>
> To further validate the effectiveness of our optimization pipeline, we performed an additional experiment: we took the Top-1 and Top-10 best-performing training shapes, encoded them into our latent space with added noise, and then optimized them using our generation and refinement pipeline. The optimized shapes achieved lower drag values compared to their original versions. This suggests that our method not only generalizes across the design space but is also capable of fine-tuning within high-performance regions.
>
> **Table 1: Comparison of Training vs. Generated Designs**
>
> | Design | Sim-Drag (Training) | Sim-Drag (Ours) |
> |--------|-------------------|------------------|
> | Top-1 | **0.2857** | 0.2944 |
> | Top-10 | **0.3017** | 0.3125 |
>
> **Table 2: Optimization Results on Best Training Designs**
>
> | Design | Sim-Drag (Before) | Sim-Drag (After) |
> |--------|------------------|------------------|
> | Top-1 | **0.2857** | 0.2912 |
> | Top-10 | 0.3017 | **0.2996** |
>
>
> >**Comment 3:"The whole point of surrogation is to amortize its training during inference. Is this the case in this paper? The paper shows the best performing shape through surrogate but what if we optimize for 1000 different drags and show the statistics? Ideally, this must be coupled with other objectives and have a multi-objective optimization."**
> >
> > **“How general is the surrogation? With the 8000 training data, can one use the trained surrogates for slightly changed conditions/applications?"**
>
> **A3:** The surrogate model in our framework serves two main purposes:
>
> - **Computational efficiency**: The surrogate model accelerates the optimization process by amortizing the training cost over multiple inference steps, reducing reliance on expensive computational solvers.
>
> - **Gradient information**: It provides differentiable gradient information, enabling efficient gradient-based optimization.
>
> Additionally, our surrogate model demonstrates a certain degree of generalization capability, extending its application beyond the training domain. To validate this, we conducted an experiment using the DrivAerNet++ dataset [1], which contains car geometries categorized into three types: **Estateback, Fastback, and Notchback**. Specifically, we retrained the surrogate models by completely excluding the **Fastback** category from training. The performance results are shown in Tabel 3.
>
> **Table 3: Surrogate Model Generalization Performance**
>
> | Surrogate Model | Relative Accuracy |
> |-----------------|-------------------|
> | UNet with Fastback | 2.87% |
> | UNet without Fastback | 7.61% |
> | GNN with Fastback | 2.09% |
> | GNN without Fastback | 9.34% |
>
> We further integrated these partially trained surrogate models into the 3DID pipeline for inverse design, the results are illustratred in Table2. These results indicate that our surrogate models maintain reasonable accuracy and yield meaningful optimization improvements even when evaluated on geometries from categories excluded during training. This observation aligns well with prior works that investigate the surrogate model's generalization capabilities[2,3].
>
> **Table 4: Pipeline Performance with Partial Surrogate Training**
>
> **UNet Guidance:**
> | Method | Pred-Drag | Sim-Drag |
> |--------|-----------|----------|
> | No guidance | 0.2964 | 0.4080 |
> | UNet (All) | 0.2733 | 0.3698 |
> | UNet (Partial) | 0.2825 | 0.3930 |
>
> **GNN Optimization:**
> | Method | Pred-Drag | Sim-Drag |
> |--------|-----------|----------|
> | No optimization | 0.2838 | 0.3958 |
> | GNN (All) | 0.2644 | 0.3628 |
> | GNN (Partial) | 0.2776 | 0.3822 |
>
> Finally, regarding multi-objective optimization, we agree that extending our surrogate-based framework to accommodate simultaneous optimization of multiple design criteria is a valuable direction, and we plan to pursue this in future work.
>
> > **Comment 4:"The fact that the physical fields are used jointly for training the encoder, does not fit the (loose) definition of physics-aware."**
>
> **A4:** We thank the reviewer for raising this important point, which is also concerned by Reviewer Vqus. In our context, "physics-aware" refers specifically to incorporating physics information in our two-stage optimization pipeline, rather than relying solely on geometry for optimizing the numerical objective. While some 2D or low-dimensional inverse design tasks can enforce physical constraints explicitly via projection or analytical formulations, this becomes significantly difficult in high-dimensional 3D inverse design problems.
>
> Instead, our framework adopts a surrogate-based strategy to integrate physics knowledge in both stages:
>
> - **In the generation phase**: We jointly embed physical fields and geometric shapes into a unified latent space, guiding the design generation toward physically plausible and performance-oriented solutions.
>
> - **In the topology optimization phase**: Our MeshGraphNet-based surrogate model takes both geometry and physical field information as input, providing gradients that guide the refinement process toward improved performance.
>
> While not enforcing physical laws explicitly, our approach offers a practical and generalizable way to incorporate physics into 3D inverse design. We will clarify this distinction in the revised manuscript.
>
> > **Comment 5: "The title is a bit too broad by not including the context (car geometries and aerodynamics)"**
>
> **A5:** While our method does not explicitly enforce physical laws, it incorporates physics via joint geometry-field encoding and surrogate-based optimization. Since the surrogate learns from data, the approach is not strictly limited to PDE-based systems. However, in this paper, we focus on aerodynamics due to the available datasets. To better reflect this scope, we decided to update the title to: **"3DID: Direct 3D Inverse Design for Aerodynamics with Physics-Aware Optimization."** We will explore broader applications in other physics-constrained domains as part of future work.
>
> > **Comment 6:"How much topology variation exist in the training data and how much the topology of optimized designs is different from the training data? Are we talking about mesh-topology or the more macro-scale car topology?"**
>
> **A6:** The DrivAerNet++ dataset[1] includes three distinct car typologies: **Notchback, Fastback, and Estateback**, representing meaningful macro-topological differences. Our generation process stays within these topological families while producing refined geometric variations, as shown in Figure 5.
>
> Our topology-preserving optimization uses Free-Form Deformation (FFD) to maintain mesh-level topology during shape refinement. We validated FFD's effectiveness by sampling 50 instances with different optimization strategies. As demonstrated in the table below, compared to directly updating the vertex position and mesh connectivity following method proposed in [4], FFD preserves watertightness and structural integrity more effectively, avoiding issues like mesh tearing or self-intersections.
>
> **Table5: Comparison of Optimization Strategies**
>
> | Method | Watertight Ratio | Self-Intersection Free Ratio |
> |--------|------------------|------------------------------|
> | FFD | 100% | 100% |
> | Continuous remeshing | 12% | 8% |
>
> > **Comment 7:"I had difficulty understanding the 2D projection part. Some clarification would be helpful to the paper."**
>
> **A7:** We thank the reviewer for noting this. The 2D projection refers to prior works that simplify the 3D inverse design task by utilizing 2D views[5] or silhouette[6] to represent the 3D geometries. These methods assume that optimizing over 2D representations will result in high-performing 3D designs, which is often inaccurate due to the loss of spatial and geometric detail. Our method avoids this limitation by working directly in 3D space. We will clarify this in the revised manuscript.
>
> > **Comment 8: "Equation 1 is hard to parse mathematically. Please check the correctness."**
>
> **A8:** We thank the reviewer for pointing out this ambiguity. We clarify that the objective function $\mathcal{J}(M)$ depends on both the geometry $M$ and its physical response $F(M)$, and will revise the equation to:
>
> $$\mathcal{J}(M) := J(M, F(M))$$
>
> This notation better reflects the functional dependency and avoids confusion with equality. We will update this in the revised version.
>
> [1] Drivaernet++: A large-scale multimodal car dataset with computational fluid dynamics simulations and deep learning benchmarks. 2024
>
> [2] Generalization capabilities of MeshGraphNets to unseen geometries for fluid dynamics. 2024
>
> [3] TripNet: Learning Large-scale High-fidelity 3D Car Aerodynamics with Triplane Networks. 2025
>
> [4] Continuous remeshing for inverse rendering. 2022
>
> [5] Surrogate modeling of car drag coefficient with depth and normal renderings. 2023
>
> [6] Interactive design of 2d car profiles with aerodynamic feedback. 2023

---

> > ### Comment · Reviewer_ht5R · 2025-08-04
> >
> > I thank the authors for their sincere effort in doing experiments in such a short time. I also thank them for answering my questions and considering my comments.
> >
> > There remain still a major and a minor point.
> > 1) The authors performed a suggested experiment where they check whether any generated design (using their method) can beat the best performing design in the training set. And the answer is no (I command them for their transparency). As I mentioned in the review, this is a sanity check. This means that the 'interpolated' designs cannot achieve better performance. In other words, the method is not capable of giving us better designs but only different designs. Why this diversity matters?
> >
> > And this begs a question: how other competing methods would fare in this evaluation?
> >
> > I don't think this point will block the paper. In fact, I think this could be one of those negative results that can increase the impact of the paper and help the community if highlighted in the paper.
> >
> > 2) My point about amortization of surrogate model was that how much we should use the surrogate in order to compensate for the effort of the training. Doing 1 optimization, 10, 10000? Suppose it is n, do we have really a scenario that we need to use the surrogate model n times?
> >
> > An additional comment: there is a work that reduces the gap between predicted (Pred) and simulated (Sim) quantities when doing surrogation, worth checking:  Autoinverse: uncertainty aware inversion of neural networks (NeurIPS 2022).

---

> > > ### Author Response · Authors · 2025-08-05
> > >
> > > We sincerely thank the reviewer for their continued engagement and thoughtful suggestions. Below, we address the remaining concerns point-by-point.
> > >
> > > > **Comment 1: "This means that the 'interpolated' designs cannot achieve better performance. In other words, the method is not capable of giving us better designs but only different designs. Why this diversity matters?"**
> > > >
> > > > **"How other competing methods would fare in this evaluation?"**
> > >
> > > **A1:** We thank the reviewer for raising this important question. While our method may not generate strictly better-performing designs under the current surrogate model, we believe the **ability to generate diverse, high-performing, and structurally valid designs from scratch** remains a significant contribution. This value stems from two key factors:
> > >
> > > 1. **From a practical perspective**, diversity is essential for real-world design workflows that involve **human-in-the-loop selection**, **and downstream constraints** (e.g., manufacturability, integration feasibility). Having a diverse pool of viable candidates enables designers to choose solutions that best fit their broader context, beyond a single performance metric.
> > >
> > > 2. **From a methodological perspective**, our approach provides a **learned, structured way to navigate the joint geometry-physics space** using a diffusion-based model.  It enables **efficient exploration within physically plausible and topologically consistent regions of the design space**. This makes it well-suited for applications in **multi-objective optimization[1], constrained design[2], which we regard as a prospective future work direction.**
> > >
> > > To address the reviewer's follow-up question on how competing methods would fare in this evaluation, we compare the top-1 and top-10 results across different optimization strategies using a shared physical-geometry representation. Additional statistical visualizations for baselines are provided in Appendix A.1.
> > >
> > > | Design | Sim-Drag(Training) | Sim-Drag(CEM) | Sim-Drag(Gradient Descent) | Sim-Drag(Ours) |
> > > | ------ | ------------------ | ------------- | -------------------------- | -------------- |
> > > | Top-1  | 0.2857             | 0.3203        | 0.3088                     | 0.2944         |
> > > | Top-10 | 0.3017             | 0.3406        | 0.3364                     | 0.3125         |
> > >
> > > > **Comment 2: "My point about amortization of surrogate model was that how much we should use the surrogate in order to compensate for the effort of the training. Doing 1 optimization, 10, 10000? Suppose it is n, do we really have a scenario that we need to use the surrogate model n times?"**
> > >
> > > **A2:** Our method is intended for scenarios where **designers need to generate and evaluate a large number of candidate geometries**, rather than optimize a single shape. Such scenarios are common in industrial design workflows, especially in **aerospace, automotive, and structural engineering**, where hundreds of alternatives must be explored. Once trained, the surrogate model supports **both efficient sampling and optimization**, enabling rapid evaluation and refinement across a broad design space.
> > >
> > > We will make this usage context more explicit in the final version to clarify when and why surrogate amortization is both practical and justified.
> > >
> > > Finally, we thank the reviewer for the helpful pointer to *AutoInverse*. We will cite this and more related work in the revised version to better situate our discussion.
> > >
> > > [1] ParetoFlow: Guided Flows in Multi-Objective Optimization. 2025
> > >
> > > [2] Diffusion Models as Constrained Samplers for Optimization with Unknown Constraints. 2024

---

> > > > ### Author Response · Authors · 2025-08-07
> > > >
> > > > Dear Reviewer ht5R,
> > > >
> > > > We sincerely appreciate your valuable engagement and thoughtful feedback.
> > > >
> > > > As the discussion phase is approaching its end, we kindly request the reviewer to let us know if the above clarifications and the previously added experiments have addressed the remaining questions. We would be happy to address any additional points the reviewer may have during the remaining time of the discussion phase.
> > > >
> > > > Thank you again for your time and consideration.
> > > >
> > > > Sincerely,
> > > >
> > > > Authors of Paper 5271

---

### Official Review · Reviewer_aPc8 · 2025-07-01

**Clarity:** 2
**Significance:** 3
**Originality:** 2
**Rating:** 4
**Confidence:** 4

**Summary:**

This paper proposes a new 3D inverse design method with physics-aware optimization, differentiating from prior work in two aspects: (1) avoiding 2D projection; (2) abstaining from fine-tuning existing 3D shapes. The 3DID framework employs a unified latent representation within a VAE to encode both shape geometry and induced physical fields. It performs two-stage generation: (1) direct latent space sampling via guided diffusion models; (2) topology-preserving mesh refinement under constraint. Results on vehicle aerodynamic shape design validate 3DID’s effectiveness.

**Questions:**

Two critical questions:

(1) should provide one more task for evaluation.

(2) should clarify the necessity of the topology preserving refinement stage.

Other questions:

(1) In Eq (2), how to formulate "constraints, such as volume, manufacturability, and boundary conditions"?

(2) In Eq (5), the = should be replaced by $\propto$

(3) What does the KL divergence mean in Eq. (3)?

(4) how does the hyperparameter $\gamma$ effect the performance? Too large $\gamma$ may make generated shape deviated from the manifold and adversarial artifacts, and too small $\gamma$  has little effect on optimizing the objective.

(5) How to pretrain the surrogate model $f_\text{GNN}$? How does it deal with out-of-distribution prediction?

(6) In Figure 4, Voxel+PCA produces non-watertight shapes. How doese 3DID guarantee watertightness?

**Ethical Concerns:**

["NO or VERY MINOR ethics concerns only"]

**Final Justification:**

I appreciate the paper and the rebuttal process. Based on its current quality and additional experiments/clarification, I would like to provide a "4: Borderline accept" rating. The reason why I do not provide a "5: Accept" rating is that its 3D experiments results are not strong enough.

**Limitations:**

yes

**Quality:**

2

**Strengths And Weaknesses:**

1. Strengths

   (1) The proposed method is novel.  The Physics-Geometry VAE unifies shape and physical field encoding in a latent space, ensuring physical consistency during design. The two-stage generation process combines objective-guided diffusion with topology-preserving refinement, addressing prior limitations.

   (2) The empirical results on vehicle aerodynamic shape optimization is significant, as shown in Table 1/2/3 and Figure 4/5.

2. Weaknesses

   (1) As a general 3D inverse design method, the current validation is limited to a single task (vehicle aerodynamics). To establish generality, at least one additional engineering task is required—e.g., 3D aircraft design, a well-studied problem in physics-based engineering design.

   (2) The motivation for the topology preserving refinement stage is ambiguously stated as "biases from training data prior," lacking clear theoretical grounding. The authors should clarify: Why is refinement needed after objective-guided sampling? A visual/intuitive illustration of the refinement process (e.g., fine-grid visualizations) is essential for understanding.

---

> ### Author Rebuttal · Authors · 2025-07-31
>
> We sincerely thank the reviewer for their constructive feedback and for recognizing the novelty of our method. Below, we provide detailed responses to each of the concerns raised.
>
> > **Comment 1: "As a general 3D inverse design method, the current validation is limited to a single task (vehicle aerodynamics). To establish generality, at least one additional engineering task is requirede.g., 3D aircraft design, a well-studied problem in physics-based engineering design."**
>
>
> **A1:** We appreciate the reviewer’s suggestion, which is also concerned by Reviewer Vqus.  Due to the lack of high-quality, open 3D aircraft simulation datasets with paired physical fields, we instead conducted additional experiments on the AhmedML dataset[1], which includes 500 diverse 3D bluff body geometries for aerodynamic evaluation.  Demonstrate that our method generalizes beyond car geometries and performs effectively across diverse shape distributions.
>
> **Table 1: Cross-Dataset Generalization Results on AhmedML**
> | |  Pred-Drag | Sim-Drag |
> |:--------------:|:-----------:|:-----------:|
> | W/O Guidance| 3.1181| 3.9549|
> | 3DID–NoTopoRefine| 2.9987| 3.7874|
> | 3DID |**2.85543**| **3.6017**|
>
>
> > **Comment 2: "The authors should clarify: Why is refinement needed after objective-guided sampling? A visual/intuitive illustration of the refinement process (e.g., fine-grid visualizations) is essential for understanding."**
>
> **A2:** The refinement stage is introduced to further improve performance beyond what can be achieved by the diffusion model alone. While the objective-guided diffusion model incorporates target-driven gradients during generation, the resulting designs remain heavily influenced by the training data’s prior distribution. To overcome this limitation, our refinement phase performs physics-aware optimization using a differentiable surrogate model. This enables fine-grained, continuous adjustments that enhance the design’s performance while preserving mesh-level topology.
>
> We agree that visualizations would greatly aid understanding. However, rebuttal guidelines do not permit the inclusion of new figures. We will provide detailed visual comparisons of pre- and post-refinement results in the revised manuscript to clearly illustrate the necessity and effectiveness of this stage.
>
> >**Comment 3: "In Eq (2), how to formulate "constraints, such as volume, manufacturability, and boundary conditions"**
>
> **A3:** We thank the reviewer for pointing out the ambiguity in the formulation of constraints in Eq. (2). To clarify this, we will revise the manuscript to explicitly state that the constrained optimization problem can be formulated as:
>
> $$M^* = \arg\min_M J(M, F(M)) \quad \text{s.t.} \quad C(M, F(M)) = 0.$$
>
> Here, M denotes the 3D shape, $F(M)$ is the induced physical field, $J$ is the design objective (e.g., drag), and $C(M,F(M))=0$ encodes geometric and physical constraints.
>
>
> > **Comment 4: "In Eq (5), the = should be replaced by $\propto$"**
>
> **A4:** We thank the reviewer for pointing out this notational issue. We will fix this in the updated version.
>
> > **Comment 5: "What does the KL divergence mean in Eq. (3)?"**
>
> **A5:** To prevent excessive variance in the physics–geometry latent representation,  the KL divergence loss is applied in the training of PG-VAE. It can be formulated as
>
> $$ \mathcal{L}_{\mathrm{KL}} = \mathrm{KL}\left( q(z \mid M, F(M)) \| \mathcal{N}(0, I) \right) .$$
>
> Where q is the posterior over the latent code, and N is the Gaussian distribution.
>
>
> > **Comment 6: "how does the hyperparameter  $\gamma$  effect the performance?"**
>
> **A6:** This is a very important question for the evaluation of our method. It is also concerned by Reviewer Vqus. We have performed additional experiments to test the influence of $\gamma$. The results are listed in the following table.  From the table, we can see that our method remains stable within a broad range of $\gamma$. As $\gamma$  increases, the guidance becomes stronger, leading to lower predicted drag and simulated drag, but if too large (e.g., $\gamma$ = 10 or 20), it causes off-manifold generations, resulting in degraded simulation performance and poor coverage. In our paper, we choose $\gamma$ based on the best evaluation performance.
>
> **Table 2: Effect of Guidance Scale on Generation Performance**
> | Guidance Method | Pred-Drag | Sim-Drag | Novelty | Coverage |
> |:-----------------:|:-----------:|:----------:|:---------:|:----------:|
> | 0               | 0.2971    | 0.3944   | 0.9177  | **0.7104** |
> | 0.1             | 0.2930    | 0.3956   | 0.9182  | 0.7101   |
> | 0.5             | 0.2812    | 0.3809   | 0.9118  | 0.7089   |
> | 1               | 0.2715    | 0.3798   | 0.9114  | 0.7027   |
> | 2               | 0.2673    | 0.3787   | 0.9167  | 0.7014   |
> | 5               | **0.2623**| **0.3766** | 0.9195  | 0.6950   |
> | 10              | 0.2823    | 0.4122   | 0.9227  | 0.5923   |
> | 20              | 0.3204    | 0.5125   | **1.4072** | 0.4512   |
>
>
> > **Comment 7: "How to pretrain the surrogate model $f_\text{GNN}$? How does it deal with out-of-distribution prediction?"**
>
>
> **A7:** We thank the reviewer for pointing out this missing detail, and we will include the full explanation in the revised version.
>
> We adopt MeshGraphNet as our surrogate model $f_\text{GNN}$, given its strong performance in mesh-based physical simulations. The surrogate is trained to predict aerodynamic drag from paired samples of geometry and ground-truth physical fields collected from the dataset. To improve robustness and mitigate out-of-distribution (OOD) issues, we adopt two strategies:
>
> 1. Noise augmentation during training by perturbing mesh node positions, which helps the model generalize better to unseen shapes.
> 2. Shape regularization during refinement optimization, which constrains the updated geometry to remain close to the training distribution.
> These strategies together ensure that both the surrogate and the final designs remain reliable and physically plausible.
>
> > **Comment 8: "In Figure 4, Voxel+PCA produces non-watertight shapes. How does 3DID guarantee watertightness?"**
>
> **A8:** Unlike voxel-based representations that operate on discrete grids and often result in artifacts such as non-watertight surfaces, 3DID leverages a continuous triplane-based latent representation that jointly encodes geometry and physical fields. This continuous encoding enables smooth and coherent surface reconstruction using the Marching Cubes algorithm. To further validate the generated geometry quality of our method, we sampled 1000 designs using our method and evaluated them for physical and geometric validity (including watertightness, self-intersection, average volume, and average surface area).  The table below compares these metrics across three stages: the original training data, our method's output  after diffusion sampling, and the final results after the refinement stage. From the table, we can see that 100% of the sampled shapes are watertight and free from self-intersections. After the optimization, we observe that the geometry completeness remains unchanged, as our topology-preserving optimization does not alter the mesh connectivity or introduce artifacts.
>
> **Table 3: Geometric Validity Assessment of Generated Automotive Designs**
> | Method | watertight Ratio  | Self-Intersection Free Ratio | Average Volume  | Average Surface Area |
> |:-----------------:|:-----------:|:----------:|:---------:|:----------:|
> | Data              | 100%    | 100%  | 5.10  | 22.27 |
> | Ours(Wo-refinement)| 100%    | 100%  | 6.10  | 24.45  |
> | Ours          | 100%    | 100%   | 5.45  | 22.56   |
>
> [1] AhmedML: High-Fidelity Computational Fluid Dynamics Dataset for Incompressible, Low-Speed Bluff Body Aerodynamics. 2024

---

> > ### Comment · Reviewer_aPc8 · 2025-08-01
> >
> > I appreciate this thorough rebuttal. It effectively resolved my questions. Therefore, my positive assessment of the paper stands and my rating remains unchanged.

---

> > > ### Author Response · Authors · 2025-08-01
> > >
> > > Thank you for your valuable feedback and continued positive assessment. We genuinely value your contributions and any further suggestions.

---

### Official Review · Reviewer_Vqus · 2025-07-02

**Clarity:** 3
**Significance:** 2
**Originality:** 2
**Rating:** 4
**Confidence:** 3

**Summary:**

The authors learn a unified latent representation that compactly encodes both the 3D shape geometry and its associated physical field (such as flow pressure). This latent space significantly reduces the dimensionality of the design problem while preserving fine geometric details and high-fidelity physics information. Building on this representation, 3DID employs a two-stage optimization pipeline: (1) a gradient-guided diffusion sampler that generates diverse candidate shapes from scratch by guiding a 3D diffusion model with the gradient of the target objective, thus biasing generation toward high-performance designs, and (2) a topology-preserving refinement stage that fine-tunes each candidate via physics-aware gradient descent (using a differentiable surrogate model) to further improve performance while maintaining the shape’s structural integrity.

**Questions:**

Demonstrating any degree of generalization or adaptability would greatly strengthen the work’s impact. The authors are requested to test two conditions: 1) Cluster Drivaernet++ using parameters and completely hold out one of the clusters as the test set. 2) Test the method on a different dataset (e.g., ShapeNet geometries) to demonstrate the degree of generalization.

Hyperparameter tuning: The paper has a complex workflow, and ablations for more key design choices are needed to understand the outcome. For example, an ablation on the effect of the objective guidance strength (the coefficient in diffusion guidance) would show sensitivity to that hyperparameter and help answer, “How does the strength of this guidance impact the downstream optimization results?” Another ablation could remove or vary the regularizers in the refinement stage (smoothness and volume preservation) to demonstrate their effect on the design outcome.

While the approach is “physics-aware” in that it uses training data from simulations, it doesn’t enforce the physical laws during generation – it relies on the surrogate’s learned approximation. If one wanted to apply 3DID to a new domain, one would have to retrain the entire model and ensure the surrogate is accurate, which could be a barrier. It will help if the authors discuss this issue.

How accurate does the encoding surrogate have to be for integration in the full pipeline?

**Ethical Concerns:**

["NO or VERY MINOR ethics concerns only"]

**Final Justification:**

I believe the author's response has addressed a few of the issues I raised regarding hyperparameters and generalizability.

**Limitations:**

A current limitation is the narrow scope of testing. A rigorous study would test generalization in multiple ways. One suggestion is cross-generalization: train the model on one subset of shapes or one objective, and test its performance on a slightly different scenario. How well does the method work on out-of-distribution geometries?

**Paper Formatting Concerns:**

The paper is well-written and organized, making a complex methodology accessible.

Ensuring consistency in terminology (e.g., sometimes “objective-guided diffusion” is referred to, elsewhere “gradient-guided diffusion”) would avoid confusion.

**Quality:**

3

**Strengths And Weaknesses:**

A core contribution is the learned latent space that jointly encodes geometry and physics (e.g., flow fields) in a compact form. The paper introduces a clever optimization strategy that balances global exploration with local refinement. The first stage, a gradient-guided diffusion sampler, leverages the generative power of diffusion models and steers them using objective gradients.

While this work is largely empirical, adding theoretical underpinnings where possible would improve its rigor. For example, providing a convergence analysis for the refinement stage (even if simplified, like proving that the FFD gradient descent will locally improve the objective under certain smoothness assumptions) would lend more credibility. Similarly, formalizing the guided diffusion by referencing known results in score-based generative modeling would address theoretical concerns. If exact analysis is intractable, even a conceptual discussion of failure modes (e.g., how too-strong guidance can lead to off-manifold samples) and how the method mitigates them (via the chosen hyperparameter) shows theoretical awareness. Grounding each algorithmic choice in either theory or established literature would dispel any impression of ad-hoc design.

Ablations to establish the impact of different hyperparameters are needed due to the complex workflow.

It is not clear how the method compares against a well-formulated optimization-only pipeline (FFD+adjoints or FFD+heuristic search), which avoids data collection, training, and the limited generalizability challenge. These methods are commonly used in aerospace and vehicle design. What are the key benefits that will convince practitioners to adopt this methodology compared to existing optimization-only approaches?

---

> ### Author Rebuttal · Authors · 2025-07-31
>
> We thank the reviewer for the detailed feedback and helpful suggestions. We are pleased that the reviewer appreciates our method's ability to balance global exploration with local refinement. Below, we address the reviewer's comments one by one.
>
> > **Comment 1: "There is no explicit enforcement of hard physical or geometric constraints during sampling."**
>
> **A1:** We thank the reviewer for this valuable suggestion. In our current pipeline, we observed that increasing the number of refinement steps does not always lead to improved performance, which we attribute to the accumulation of surrogate bias during optimization. To mitigate this issue, we introduce two regularization terms that help constrain the optimization within the data manifold and reduce the risk of generating out-of-distribution designs. As shown in our response to **Comment 3**, these regularizers are empirically effective in stabilizing the refinement process. We acknowledge this as a limitation of our method and plan to explore more principled convergence strategies in future work.
>
>
> > **Comment 2: "Formalizing the guided diffusion by referencing known results in score-based generative modeling would address theoretical concerns."**
>
> **A2:** In our method, we adjust the predicted noise at each denoising step by injecting the gradient of the task objective $J$
>
> $$\epsilon \prime_\phi(z_t, t) = \epsilon_\phi(z_t, t) + \gamma \nabla_{z_t} J(\hat{z}_0(z_t)).$$
>
> This objective-aware guidance can be interpreted as replacing the unconditional score $\nabla_{z_t} \log p(z_t)$ in score-based models with the conditional score[1]:
>
> $$\nabla_{z_t} \log p(z_t | J) = \nabla_{z_t} \log p(z_t) + \nabla_{z_t} \log p(J | z_t),$$
>
> where the second term is approximated by $-\nabla_{z_t} J(\hat{z}_0(z_t))$. This aligns with prior work viewing the reverse diffusion process as a gradient-based optimization over the data distribution. Thus, our method can be understood as performing optimization over a composite log-likelihood objective $\log p(z_t | J)$, integrating both data fidelity and design performance into a unified sampling process. We will incorporate this theoretical clarification into the revised version.
>
> > **Comment 3: "A conceptual discussion of failure modes and how the method mitigates them shows theoretical awareness."**
> > **Ablations for key hyperparameters are needed due to the complex workflow.**
>
> **A3:** The influence of hyperparameters is also concerned by Reviewer aPc8. We address this concern through two key ablation studies.
>
> **1. Ablation on Guidance Scale ($\gamma$):**
>
> We evaluate how varying the guidance strength $\gamma$ would affects generation quality. As shown in the Table 1, increasing $\gamma$ initially improves performance by enhancing the effect of the target-driven gradient, resulting in lower predicted and simulated drag. However, excessively large $\gamma$ (e.g., 10 or 20) causes the generation to drift off the learned data manifold, leading to worse simulation performance and a sharp drop in coverage. In our paper, we choose $\gamma$ based on the best evaluation performance.
>
> **Table 1: Effect of Guidance Scale on Generation Performance**
>
> | Guidance Scale ($\gamma$) | Pred-Drag | Sim-Drag | Novelty | Coverage |
> |:-------------------:|:-----------:|:----------:|:---------:|:----------:|
> | 0 (no guidance) | 0.2971 | 0.3944 | 0.9177 | **0.7104** |
> | 0.1 | 0.2930 | 0.3956 | 0.9182 | 0.7101 |
> | 0.5 | 0.2812 | 0.3809 | 0.9118 | 0.7089 |
> | 1 | 0.2715 | 0.3798 | 0.9114 | 0.7027 |
> | 2 | 0.2673 | 0.3787 | 0.9167 | 0.7014 |
> | 5 | **0.2623** | **0.3766** | 0.9195 | 0.6950 |
> | 10 | 0.2823 | 0.4122 | 0.9227 | 0.5923 |
> | 20 | 0.3204 | 0.5125 | **1.4072** | 0.4512 |
>
> **2. Ablation on Refinement Regularizers:**
>
> We conducted additional experiments to assess the effect of regularization terms in the topology preserving refinement stage, namely volume preservation and control point smoothness. As shown in the Table 2, removing the volume constraint results in a noticeable drop in performance, particularly in simulated drag and coverage, due to uncontrolled volume shrinkage during refinement. Removing the smoothness regularizer leads to degraded drag performance, indicating that the lack of regularization results in less effective shape refinement.
>
> **Table 2: Effect of Regularization Terms on Refinement Performance**
>
> | Method | Pred-Drag | Sim-Drag | Novelty | Coverage | Volume |
> |--------|:-----------:|:----------:|:---------:|:----------:|:---------:|
> | With both regularizers | 0.2607 | 0.3536 | 1.1709 | 0.4300 | 5.9578 |
> | Without volume | 0.2765 | 0.4190 | 1.3472 | 0.2375 | 4.7624 |
> | Without smooth regularizer | 0.2715 | 0.4070 | 1.2762 | 0.3711 | 5.4654 |
>
> > **Comment 4: "It is not clear how the method compares against optimization-only pipelines such as FFD+adjoints or FFD+heuristic search."**
> > **"What are the key benefits that will convince practitioners to adopt this methodology compared to existing optimization-only approaches?"**
>
> **A4:** Compared to traditional optimization-only pipelines such as FFD+adjoints or FFD+heuristic search, our method offers two key advantages. First, our diffusion-based sampling stage enables automatic exploration of diverse and high-quality initial designs, removing the need for manually selected initial shapes, which are often critical in conventional pipelines. Second, while our refinement stage shares similarities with methods like FFD+adjoints in using gradient-based updates, it leverages a pre-trained surrogate model, allowing efficient optimization without requiring access to differentiable solvers.
>
> > **Comment 5: "Demonstrating any degree of generalization or adaptability would greatly strengthen the work's impact."**
> >
> > **"Please test generalization by (1) Cluster DrivAerNet++ using parameters and completely hold out one of the clusters as the test set (2) evaluating on a different dataset"**
> >
> > **"How accurate does the encoding surrogate have to be for integration in the full pipeline?"**
>
> **A5:** We thank the reviewer for raising these important questions regarding generalization. We address them from two perspectives:
>
> **(1) Generalization of our pipeline**
> To evaluate whether our method generalizes beyond car geometries, we conducted additional experiments on the **AhmedML** dataset[2], which contains 500 diverse 3D bluff body geometries with simulated aerodynamic fields. This dataset differs significantly from DrivAerNet++[3] in shape distribution. As illustrated in Table 3, our method still demonstrates effective performance, indicating that our pipeline is not overfitted to a specific dataset and can generalize to new shape families.
>
> **Table 3: Cross-Dataset Generalization Results on AhmedML**
>
> | Method | Pred-Drag | Sim-Drag |
> |--------|:-----------:|:----------:|
> | W/O Guidance | 3.1181 | 3.9549 |
> | 3DID–NoTopoRefine | 2.9987 | 3.7874 |
> | 3DID | **2.8554** | **3.6017** |
>
> **(2) Generalization of our surrogate model**
>
> To evaluate the generalization of our surrogate model, we retrained it on the DrivAerNet++[3] dataset while holding out the Fastback category. We then integrated the partially trained model into our 3DID pipeline. As shown in Table 4, although the guidance is weaker than with full-data training, it still improves performance over unguided generation.
>
> **Table 4: Pipeline Performance with Partial Surrogate Training**
>
> **UNet Guidance:**
> | Method | Pred-Drag | Sim-Drag |
> |--------|:-----------:|:----------:|
> | No guidance | 0.2964 | 0.4080 |
> | UNet (All) | 0.2733 | 0.3698 |
> | UNet (Partial) | 0.2825 | 0.3930 |
>
> **GNN Optimization:**
> | Method | Pred-Drag | Sim-Drag |
> |--------|:-----------:|:----------:|
> | No optimization | 0.2838 | 0.3958 |
> | GNN (All) | 0.2644 | 0.3628 |
> | GNN (Partial) | 0.2776 | 0.3822 |
>
>
> > **Comment 6: "The method is physics-aware only through training data; it doesn't enforce physical laws during generation."**
> > **"The method relies on a surrogate trained from simulations without enforcing physical laws, limiting generalization and requiring retraining in new domains."**
>
> **A6:** We thank the reviewer for highlighting this important point, which is also raised by Reviewer ht5R. In our context, **"physics-aware"** refers to incorporating physics information into both stages of our optimization pipeline, rather than relying solely on geometry for guiding the design process. While explicit enforcement of physical laws via projection operators is feasible in some 2D or low-dimensional inverse design tasks, such formulations become intractable in high-dimensional 3D settings.
>
> Instead, our framework adopts a surrogate-based strategy to integrate physics knowledge in both stages:
>
> - **In the generation phase**: We jointly embed physical fields and geometric shapes into a unified latent space, guiding the design generation toward physically plausible and performance-oriented solutions.
>
> - **In the topology optimization phase**: Our MeshGraphNet-based surrogate model takes both geometry and physical field information as input, providing gradients that guide the refinement process toward improved performance.
>
> While this does not enforce physical laws in the strict sense, it enables a scalable and practical way to incorporate physics into 3D inverse design.
>
> [1] Score-Based Generative Modeling through Stochastic Differential Equations. 2021
>
> [2] AhmedML: High-Fidelity Computational Fluid Dynamics Dataset for Incompressible, Low-Speed Bluff Body Aerodynamics. 2024
>
> [3] Drivaernet++: A large-scale multimodal car dataset with computational fluid dynamics simulations and deep learning benchmarks. 2024

---

> > ### Comment · Reviewer_Vqus · 2025-08-03
> >
> > Thank you for the author's response. I believe it has addressed a few of the questions I requested. I am still concerned about a lack of comparisons with optimization baselines (one using surrogate or adjoint) and hope the work considers these comparisons.

---

> > > ### Author Response · Authors · 2025-08-05
> > >
> > > > **Comment: Thank you for the author's response. I believe it has addressed a few of the questions I requested. I am still concerned about a lack of comparisons with optimization baselines (one using surrogate or adjoint) and hope the work considers these comparisons.**
> > >
> > > We thank the reviewer for the continued engagement and helpful follow-up suggestions.
> > >
> > > Due to the high computational cost associated with running full CFD-based simulations in optimization and obtaining adjoint gradients, we **adopt a differentiable surrogate model as a proxy for the physical solver**. This surrogate enables efficient gradient-based optimization, and under this setting, our second-stage refinement can effectively be regarded as a **surrogate-based adjoint method**.
> > >
> > > To compare against different FFD-based optimization baselines, we conducted experiments using the **Cross-Entropy Method (CEM)** as a representative heuristic optimizer. Since such methods are highly sensitive to initialization, we initialized both the **surrogate+heuristic search** and **surrogate+adjoint** baselines using samples generated by our diffusion model **without any guidance**. We also compare against our **full two-stage pipeline**, in which the initial designs are generated via **objective-guided diffusion** and then refined using surrogate gradients.
> > >
> > > The results are demonstrated in the table below. It can be seen that our full pipeline outperforms both optimization strategies for both predicted drag and simulation drag. Notably, solely using the surrogate+adjoint method fails to reach comparable performance, emphasizing the importance of initialization. In contrast, our full pipeline leverages objective-guided diffusion to generate high-quality initial designs, which significantly improves the effectiveness of the subsequent refinement stage.
> > >
> > > | Method                           | Pred-Drag | Sim-Drag |
> > > | -------------------------------- | --------- | -------- |
> > > | Surrogate+adjoint                | 0.2764    | 0.3757   |
> > > | Surrogate+heuristic search (CEM) | 0.2786    | 0.3829   |
> > > | Ours                             | 0.2607    | 0.3536   |

---

> > > > ### Author Response · Authors · 2025-08-07
> > > >
> > > > Dear Reviewer Vqus,
> > > >
> > > > We sincerely appreciate your valuable engagement and thoughtful feedback.
> > > >
> > > > As the discussion phase is approaching its end, we kindly request the reviewer to let us know if the above clarifications and the previously added experiments have addressed the remaining questions. We would be happy to address any additional points the reviewer may have during the remaining time of the discussion phase.
> > > >
> > > > Thank you again for your time and consideration.
> > > >
> > > > Sincerely,
> > > >
> > > > Authors of Paper 5271

---

### Official Review · Reviewer_egpt · 2025-07-03

**Clarity:** 3
**Significance:** 3
**Originality:** 2
**Rating:** 5
**Confidence:** 1

**Summary:**

The paper introduces 3DID, a framework for direct 3D inverse design that addresses the limitations of prior approaches constrained by 2D projections or initial geometry assumptions. Instead of operating in the raw high-dimensional space, 3DID first learns a compact, unified latent representation that couples geometry and physics through a triplane-based VAE. It then performs a two-stage optimization: (1) a gradient-guided diffusion sampler that generates diverse candidate designs in latent space with objective-aware steering, and (2) a topology-preserving refinement stage based on free-form deformation that further improves the design while maintaining geometric integrity. Applied to aerodynamic shape optimization, 3DID outperforms existing methods in objective quality, novelty, and simulation fidelity.

**Questions:**

While the diffusion process is guided by gradients of the objective, there is no explicit enforcement of hard physical or geometric constraints during sampling. This raises the question of how often does the sampling stage produce geometries that are physically invalid or impractical?

**Ethical Concerns:**

["NO or VERY MINOR ethics concerns only"]

**Final Justification:**

Based on the overall quality of the submission and the authors’ responses to my comments in the rebuttal, I’m inclined to recommend acceptance for this paper.

**Limitations:**

Yes.

**Paper Formatting Concerns:**

No concerns

**Quality:**

3

**Strengths And Weaknesses:**

The methodology is well presented and evaluated. Results on aerodynamic shape optimization show improvements in simulated drag and design novelty over baselines. The proposed physics-geometry triplane embedding and the gradient-guided diffusion and topology-preserving refinement look both novel.
Clarity is generally good, with clear figures. The work targets a challenging and relevant problem in scientific design and makes clear progress compared to existing methods. Overall, I find this to be a solid and well-executed paper with meaningful contributions, and I lean toward acceptance.

---

> ### Author Rebuttal · Authors · 2025-07-31
>
> We thank the reviewer for the positive feedback. We appreciate that the reviewer thinks our work is novel in the method and makes clear progress compared to existing methods. Below, we address the reviewer's concerns one by one.
>
> > **Comment 1: "There is no explicit enforcement of hard physical or geometric constraints during sampling."**
>
> **A1:** We sincerely thank the reviewer for pointing out this important aspect, which indeed opens a valuable direction for future work. However, we would like to respectfully note that the gradient-guided mechanism already helps steer the generative process toward regions of the design space that implicitly satisfy geometric constraints and exhibit improved physical performance. Specifically, for **geometric constraints**, our diffusion model is trained on a large amount of automotive shapes, which enables it to learn the underlying structure and generate topologically coherent geometries (see Comment 2). For **physical constraints**, since inverse design lacks direct PDE-based formulations as in forward simulations, we instead rely on gradient guidance from the performance predictor to bias sampling toward designs with better aerodynamic properties. This soft guidance effectively incorporates physical awareness without requiring explicit constraint formulations.
>
> > **Comment 2: "How often does the sampling stage produce geometries that are physically invalid or impractical?"**
>
> **A2:** Thanks for suggesting such an important experiment. To validate the plausibility of the generated geometries, we sampled 500 designs using our method and evaluated them for physical and geometric validity (including watertightness, self-intersection, average volume, and average surface area). The table below compares these metrics across three stages: the original training data, our method's output after sampling, and the final results after the refinement stage. From the table, we can see that 100% of the sampled shapes are watertight and free from self-intersections. This robustness is largely attributed to the proposed continuous latent embedding, which, when combined with the marching cubes algorithm, ensures that the decoded shapes remain topologically coherent and geometrically well-formed. After the optimization, we observe that the geometry completeness remains unchanged, as our topology-preserving optimization does not alter the mesh connectivity or introduce artifacts.
>
> **Table 1: Geometric Validity Assessment of Generated Automotive Designs**
>
> | Method                | Watertight Ratio | Self-Intersection Free Ratio | Average Volume | Average Surface Area |
> | --------------------| :----------------: | :----------------------------:| :--------------: | :--------------------: |
> | Data                  | 100%             | 100%                         | 5.10           | 22.27                |
> | Ours (w/o refinement) | 100%             | 100%                         | 6.10           | 24.45                |
> | Ours                  | 100%             | 100%                         | 5.45           | 22.56                |

---

> > ### Author Response · Authors · 2025-08-05
> >
> > We thank the reviewer for engaging in the discussion. We are eager to know whether the additional experiments and clarifications we provided have addressed the reviewer’s concerns. If there are any further questions during the remaining time of the discussion phase, we would be more than happy to respond.

---

### Author Response · Authors · 2025-08-09
**General Response**

We thank all reviewers for their thorough and constructive feedback. We are pleased that the reviewers recognize the novelty of our method, the clarity of our presentation, and our contribution to tackling a challenging and underrepresented problem. Reviewers also highlighted that our framework achieves convincing experimental results and makes clear progress compared to existing methods (Reviewers egpt, aPc8). Based on the reviewers’ valuable feedback, we have conducted additional experiments, analyses, and clarifications during the rebuttal and discussion phase, and will incorporate these updates into the final version of the paper. The major additional experiments and improvements are as follows:

1. **Validating geometric robustness (Reviewers egpt, aPc8)**
   We quantitatively evaluated 500 generated designs, confirming that 100% are watertight and free from self-intersections, thanks to our continuous latent representation and topology-preserving optimization.
2. **Demonstrating generalization capability (Reviewers Vqus, aPc8)**
   We tested our method on the AhmedML dataset [1] (bluff-body aerodynamics) and under cross-category generalization on DrivAerNet++ [2] by holding out the Fastback class during surrogate training. In both cases, our pipeline demonstrated strong adaptability and maintained performance gains, indicating the generalization capability of our method.
3. **Analyzing hyperparameters and regularizers (Reviewers Vqus, aPc8, ht5R)**
   We conducted detailed ablations on the guidance scale in the diffusion stage and on the volume/smoothness regularizers in the topology-preserving refinement stage. Results show that moderate guidance improves performance, while excessive guidance causes off-manifold drift. Both regularizers play an important role in ensuring stable optimization and strong final performance.
4. **Clarifying “physics-aware” definition (Reviewers Vqus, ht5R)**
   In our context, “physics-aware” refers to jointly embedding physical fields and geometry in the latent space and using surrogate gradients in the refinement stage, rather than relying solely on geometry to optimize the numerical objective. This approach offers a practical alternative to explicit physical constraint enforcement, which is often intractable in complex 3D inverse design tasks.
5. **Comparing with optimization-only baselines (Reviewer Vqus)**
   We implemented surrogate+adjoint and surrogate+heuristic search (CEM) baselines, and under a fair comparison, our full two-stage method achieved superior performance in both predicted and simulated drag, highlighting the critical role of guided diffusion in generating high-quality initial designs.
6. **Explaining refinement necessity (Reviewer aPc8)**
   We clarified that refinement mitigates the bias from training priors and enables fine-grained, physics-aware optimization beyond the capabilities of diffusion generation alone.
7. **Clarifying the value of design diversity (Reviewer ht5R)**
   While not surpassing the single best training design, our method generates a diverse set of high-performing, physically valid designs, enabling flexibility for human-in-the-loop selection and downstream constraints.

[1] AhmedML: High-Fidelity Computational Fluid Dynamics Dataset for Incompressible, Low-Speed Bluff Body Aerodynamics. 2024

[2] Drivaernet++: A large-scale multimodal car dataset with computational fluid dynamics simulations and deep learning benchmarks. 2024

---

### Decision · Program_Chairs · 2025-09-17

**Decision:**

Accept (poster)

**Comment:**

This paper presents 3DID, a framework for direct 3D inverse design that learns a joint physics–geometry latent representation and performs a two-stage optimization with gradient-guided diffusion sampling and topology-preserving refinement. The method addresses the limitations of prior 2D or fine-tuning-based approaches and achieves strong results on aerodynamic shape optimization, validated with both surrogate predictions and ground-truth simulations. Reviewers raised questions on constraint handling, generalization, refinement necessity, and baseline comparisons, but the authors provided thorough clarifications and additional experiments, including ablations, cross-dataset tests, and optimization-only baselines. The paper is technically solid, clearly presented, and tackles a difficult and underrepresented problem with meaningful contributions. Overall, this is a clear accept.